# Optimal Design of Array Coils for Multi-Target Adjustable Electromagnetic Brain Stimulation System

**DOI:** 10.3390/bioengineering10050568

**Published:** 2023-05-09

**Authors:** Tingyu Wang, Lele Yan, Xinsheng Yang, Duyan Geng, Guizhi Xu, Alan Wang

**Affiliations:** 1School of Electrical Engineering, Hebei University of Technology, Tianjin 300130, China; 2State Key Laboratory of Reliability and Intelligence of Electrical Equipment, Hebei University of Technology, Tianjin 300130, China; 3Auckland Bioengineering Institute, University of Auckland, Auckland 1010, New Zealand; 4Centre for Brain Research, Faculty of Medical and Health Sciences, University of Auckland, Auckland 1010, New Zealand; 5Centre for Medical Imaging, Faculty of Medical and Health Sciences, University of Auckland, Auckland 1010, New Zealand

**Keywords:** temporal interference, transcranial magnetic stimulation, deep brain stimulation, array coils, electric field

## Abstract

Temporal interference magnetic stimulation is a novel noninvasive deep brain neuromodulation technology that can solve the problem of balance between focus area and stimulation depth. However, at present, the stimulation target of this technology is relatively single, and it is difficult to realize the coordinated stimulation of multiple brain regions, which limits its application in the modulation of multiple nodes in the brain network. This paper first proposes a multi-target temporal interference magnetic stimulation system with array coils. The array coils are composed of seven coil units with an outer radius of 25 mm, and the spacing between coil units is 2 mm. Secondly, models of human tissue fluid and the human brain sphere are established. Finally, the relationship between the movement of the focus area and the amplitude ratio of the difference frequency excitation sources under time interference is discussed. The results show that in the case of a ratio of 1:5, the peak position of the amplitude modulation intensity of the induced electric field has moved 45 mm; that is, the movement of the focus area is related to the amplitude ratio of the difference frequency excitation sources. The conclusion is that multi-target temporal interference magnetic stimulation with array coils can simultaneously stimulate multiple network nodes in the brain region; rough positioning can be performed by controlling the conduction of different coils, fine-tuning the position by changing the current ratio of the conduction coils, and realizing accurate stimulation of multiple targets in the brain area.

## 1. Introduction

Brain neuromodulation can be divided into two types: invasive and noninvasive, according to the mode of operation. Invasive neuromodulation includes deep brain stimulation (DBS) and optogenetics [1,2]. It has the advantages of good focus and fast response but requires precise surgical procedures to embed electrodes and related electrical stimulation devices in the body, accompanied by certain surgical risks and damage. Noninvasive neuromodulation is a noninvasive intervention on the brain using physical factors such as electricity, magnetism, sound, light, heat, etc. The methods of noninvasive regulation of brain neural activity mainly include transcranial magnetic stimulation (TMS), transcranial electrical stimulation (TES), transcranial ultrasound stimulation (TUS), etc. [3].

Among them, the principle of transcranial magnetic stimulation technology is to place the pulsed magnetic field generated by the energized coil above the scalp, through the skull, to reach the targeted brain area and generate an induced electric field, which depolarizes neurons and generates an action potential, thus affecting the metabolism and electrical activity of nerves in the brain [4,5,6,7].

However, the stimulation effect of transcranial magnetic stimulation will be limited by the tradeoff between the focus of the target and the depth of the induced electric field. For example, a larger-sized coil can achieve a deeper intracranial penetration distance and provide a stronger induced electric field, but the stimulation area is larger; while the smaller size coil can provide a narrower stimulation area, but the stimulation depth is not great enough. To sum up, to stimulate the deep intracranial area, a solution that can take into account the intensity of the induced electric field, the depth of the stimulation, and the focus of the target is urgently needed [8].

According to references [9,10,11,12,13,14,15], both temporal interference transcranial magnetic stimulation and temporal interference transcranial electrical stimulation are deep brain stimulation schemes based on the principle of temporal interference. Adjusting the spatial position of the coil can generate a low-frequency envelope-induced superimposed electric field in the target area, making neurons follow this low-frequency envelope oscillation to generate electrical activity. In non-target areas, superimposed induction electric fields are dominated by high-frequency induction electric fields, and combined with the extensively accepted low-pass filtering characteristic mechanism of neuronal membranes, the neural electrical activity in this area will not be able to follow very-high-frequency oscillating electric fields [13]. Grossman et al. conducted deep brain electrical stimulation on mice in 2017, using time interference electrical stimulation technology. Two pairs of electrodes were fed with currents at frequencies of 2000 Hz and 2010 Hz, with a frequency difference of 10 Hz. They also applied anti-phasic current drive technology to eliminate crosstalk between the two current sources. The results showed that the 10 Hz differential frequency envelope could cause nerve discharges synchronized with the envelope. At the same time, the team successfully recorded the responses of neurons at different depths in the mouse hippocampus through patch clamp technology. In addition, the team also studied the effect of the current amplitude ratio of the two groups of electrodes on the focus area [9]. Majid Memarian Sorkhabi et al. conducted a simulation analysis on the low-frequency envelope induction electric field of temporal interference transcranial magnetic stimulation [12]. The results showed that compared with traditional transcranial magnetic stimulation, magnetic stimulation combined with temporal interference has a deeper stimulation distance and a focusing region with a smaller area; the movement of the target point is related to the amplitude ratio of the two excitation sources, which is similar to the movement law of the target point of time interference transcranial electrical stimulation. The neurons in the H-H model could generate action potentials following the low-frequency envelope of the temporal interference transcranial magnetic stimulation. Zonghao Xin et al. proposed a four-leaf clover two-layer coil model based on the research of Majid Memarian Sorkhabi, which can increase the induced electric field intensity under the time interference low-frequency envelope and optimize the target focus area [13]. Adam Khalifa et al. designed and optimized a custom solenoid capable of generating temporally interfering induced electric fields to stimulate the rodent brain and express neurons for tracing through the C-Fos activation situation [14]. The results showed no C-Fos expression in the region affected by only one high-frequency magnetic field, indicating that the recruitment of neural activity in the non-target region was ineffective. In contrast, the region subjected to the superimposed low-frequency envelope-induced field showed stronger C-Fos expression, indicating that temporal interference TMS could activate neuronal action potentials. In 1986, Greenberg et al. first discovered the expression of C-Fos within a few minutes after activating nerve cells. Utilizing this characteristic, C-Fos can serve as a marker of neuronal activity; it is typically used for gene expression in electrical and magnetic stimulation of nerve cells at present [15].

In addition, in recording neural activity and monitoring the focal area of deep brain stimulation, S. Beatriz Goncalves et al. proposed a single LED phototube neural tool, which is internally integrated with RTD (resistance temperature detector) for sensing. It can evaluate tissue temperature over a period of time and perform thermal imaging of brain tissue near the stimulus focus. RTD’s average accuracy is 0.2 °C at a normal body temperature of 37 °C and takes into account the ability of electrical recording and light stimulation at the same time. The RTD thin film is integrated into the silicon carbon needle to adapt to light stimulation, electrophysiological recording point, temperature sensing, etc. This electrode has great potential in recording and stimulating neural activity [16]. Tiago Matheus Nordi et al. proposed a biopotential acquisition system that can be used in deep brain stimulation. The core component is a low-noise amplifier (LNA), which is designed in a voltage/low-power CMOS process, and its area is only 122 μm × 283 μm. The test results show that the gain is 38.6 dB, the bandwidth is −3 dB, the frequency is 2.3 kHz, and the power consumption is 2.8 μW. It is expected to be applied in deep brain stimulation in the future [17].

The above studies have proved that TMS based on temporal interference can selectively stimulate neurons in deep brain regions and has good stimulation depth and focus. However, there are still problems, such as a single stimulation target. It is challenging to achieve multi-brain region multi-target co-stimulation, which limits its application in the modulation of multiple nodes in the brain network. The brain network that forms the basis of brain function sometimes contains multiple nodes that help regulate the function of the brain network, and multiple nodes need to be stimulated simultaneously. In addition, some brain networks involve nodes deep in the brain, such as the hippocampus in memory networks. Therefore, it is necessary to develop new technologies that can simultaneously activate multiple targets in deep brain regions to modulate the function of brain networks.

This paper introduces the concept of multi-brain area co-stimulation, a new brain stimulation technique designed to simultaneously stimulate multiple nodes of a brain network in deep brain regions. Specifically, a multi-coil array structure was designed, which can drive the conduction of different coils to simultaneously stimulate multiple targets through an algorithm [18,19,20,21,22]. Moreover, rough positioning can be performed by controlling the conduction of different coils; fine-tuning of the position occurs by adjusting the current ratio of the conduction coils to achieve precise stimulation of multiple brain regions and targets. It is expected to be used for synergistic stimulation of multiple brain regions to treat different diseases and regulate brain network functions precisely.

## 2. Materials and Methods

### 2.1. Multi-Target Temporal Interference Magnetic Stimulation Model Characterization Setup

The multi-target temporal interference system based on array coils is shown in Figure 1, which consists of a power module, DC/AC module, control module, and array coils. The high-frequency LCC resonant circuit constitutes a single-phase multi-channel charging structure, which has the advantage of constant current and fast charging [23]. Capacitors and converters form a cell, which can control the stimulation mode of one coil. By cascading multiple cells, the number of coils can be expanded from 1 to n, thereby realizing multi-target temporal interference of array coils. The control terminal of this system can select the opening of a certain coil unit. In addition, it can transmit control commands such as frequency and amplitude to the discharge module and can also monitor the charging voltage of the DC capacitor and the stimulating current of the stimulating coil in real time to prevent overvoltage and overcurrent [24,25,26,27,28]. In this paper, through the COMSOL simulation verification, multi-target synergistic stimulation can be realized by turning on multiple sets of coils simultaneously; changing the coils’ current ratio can realize fine-tuning of the stimulation targets without moving the coil position.

To establish a temporal interference magnetic stimulation model, at least two coils are required for stimulation, and the low-frequency envelope modulation waveform in Figure 2 is superimposed, where $AM_{n}$ represents the amplitude modulation intensity, which is defined by Equation (1). In this study, seven circular coils are used to construct a multi-target coil array, numbered 1, 2, 3, 4, 5, 6, and 7, respectively. All coils were circular coils of the same size, arranged as shown in Figure 3A,B. The center of the circular coil numbered 1 coincides with the center of the array, which is called the central coil; the remaining numbered coils (from small to large) are arranged clockwise around the central coil. The selection of radius has two factors: firstly, considering that the diameter of the five-layer sphere model is 184 mm, the head circumference of adults is 540–580 mm, and there are at least three circles in the same column. Secondly, the coil radius is approximately positively correlated with the stimulus intensity. Specifically, as the coil radius increases, the cortical stimulus intensity gradually increases. Thus, the coil unit is selected with an outer radius of 25 mm [29]. The wire diameter is 2 mm, the number of turns is 6, and the material is copper with good conductivity. The shell is made of an insulating epoxy resin board, which is characterized by good mechanical properties under medium temperature conditions, stable electrical properties under high humidity conditions, good heat resistance and moisture resistance, and can protect personnel safety. On the one hand, referring to the figure-eight coil, only when coil 1 and coil 2 are fed with reverse current an induced electric field focusing area will be generated at their intersection [30,31,32,33]. Similarly, when coil 1 and coils 2, 3, and 6 are powered on, there are three focal areas where the induced electric field can be generated near coil 1. On the other hand, when coil 1 is fed with alternating current with a frequency of 1.00 KHz, when coils 2, 3, and 6 are fed with 1.05 KHz alternating current, a low frequency modulated at 50 Hz will be generated at the three focal points envelope-induced electric field. Due to the low-pass mechanism of neuronal membranes, high-frequency electric fields will not activate neurons, and effective stimulation can only be achieved in areas where interference occurs, and low-frequency envelopes are superimposed. Compared with traditional magnetic stimulation, the focus is greatly improved. To quantify the low-frequency envelope-induced electric field amplitude modulation intensity $AM_{n}$ at any point, the following Equation (1) was used
(1)$$AM_{n}=\left|\left({\bar{E}}_{n1}+{\bar{E}}_{n2}\right)-|{\bar{E}}_{n1}-{\bar{E}}_{n2}|\right|,$$
where represents the amplitude modulation intensity of the low-frequency envelope-induced electric field at any position *n* in space, ${\bar{E}}_{n1}$ represents the 1 KHz induced electric field modulus at position *n*, and ${\bar{E}}_{n2}$ represents the 1.05 KHz induced electric field modulus at position *n*.

According to the TMS and PET research experience provided in literature [34,35], brain regions activated by the TMS have significantly higher E-field components parallel to pyramidal neurons in the dendrite-to-axon orientation and in the tangential direction (i.e., parallel to interneurons) at a high gradient. This article assumes that the axis of the target nerve cell is parallel to the x-component of the induced electric field (that is, the tangential direction of coil 1, coil 2, and coil 1, coil 5), so it is only necessary to calculate the ${\bar{E}}_{nx}$ of the x-component of the induced electric field at any position n. Therefore, the above equation is corrected as (2)
(2)$$AM_{nx}=\left|\left({\bar{E}}_{n1x}+{\bar{E}}_{n2x}\right)-|{\bar{E}}_{n1x}-{\bar{E}}_{n2x}|\right|,$$
where $AM_{nx}$ is the amplitude modulation intensity of the x-component of the low-frequency envelope-induced electric field at any position *n* in space, ${\bar{E}}_{n1x}\;$ represents the modulus of the x-component of the induced electric field at 1 KHz at position *n*, and ${\bar{E}}_{n2x}$ represents the modulus of the x-component of the induced electric field at 1.05 KHz at position *n*.

### 2.2. Finite Element Modeling

There are two main tasks for simulating multi-target temporal interference magnetic stimulation: (1) Implementation of multi-target array coil temporal interference. This paper conducted temporal interference simulations of two sets of physiological saline models for multi-target array coils, with the first set being conducting coils 1, 2, and 5. The second group is the conduction coils 1, 5, and 6. In addition, in the ball model, the distribution of the focusing area was analyzed when coils 1, 2, and 5 were turned on. (2) Analyze the relationship between the current ratio of two different frequency coils and the movement of the focusing area. Apply a 1 KHz AC signal to coil 1 and a 1.05 KHz AC signal to coil 2, adjust their current ratio, and observe the changes in the focusing area.

All simulations are carried out on the simulation platform COMSOL, a multi-physics coupling finite element simulation software that includes multiple modules such as mechanics, electromagnetics, acoustics, and heat transfer. For the simulation of temporal interference transcranial magnetic stimulation, the electromagnetic field module of COMSOL is selected for finite element analysis. The specific steps are: (1) Two physical models are built, one of which is the temporal interference multi-target magnetic stimulation saline model, which is used to simulate human tissue fluid-induced electric field distribution, including a cuboid with a size of 300 × 300 × 40 mm and an array coil. The reasons for choosing this cuboid size are: firstly, if the size is too large, it will occupy more computing resources and prolong the calculation time; secondly, the model size should not be smaller than the peripheral size of the array coil to prevent the induced electric field distribution area from not representing the spatial distribution of the electric field of the array coil; finally, considering that deep brain stimulation usually occurs in the white matter and gray matter regions within the range of 15–35 mm of the brain. Based on the above reasons, the size positioning selected was 300 × 300 × 40 mm [36,37,38]. The other is a temporal interference multi-target magnetic stimulation sphere model, which has five layers, including scalp, skull, cerebrospinal fluid (csf), and gray matter, and can be used to simulate the induced electric field distribution in the human brain. The scalp, skull, cerebrospinal fluid (csf), and gray matter were 4.9 mm, 7.4 mm, 2.5 mm, and 7.2 mm, respectively. The innermost white matter of the brain was a sphere with a radius of 70 mm [39,40]. Here, it should be emphasized that the reason for choosing the five-layer sphere model instead of the real human brain model in this article is because the concentric sphere model has been used in multiple studies [9,14,31,41] with high accuracy and little difference from the real head model. It can be used for qualitative and quantitative analysis to a certain extent. Secondly, using a real brain model, such as MRI segmentation of images, will generate a large number of units. Although many articles on coil optimization design and analysis of induced electric field distribution use real brain models for simulation, transient simulations with small steps, such as time interference (usually within 0.1 ms), consume more computer resources. (2) Assign material properties. The conductivity of the saline model was set to 0.333 S/m, and the relative permittivity was 100; the conductivity of the white matter was set to 0.062 S/m, and the relative permittivity was 69,811; the conductivity of the gray matter was set to 0.098 S/m, relative permittivity was 69,811; cerebrospinal fluid conductivity was set to 2 S/m, relative permittivity was 109; skull conductivity was set to 0.021 S/m, relative permittivity was 2702; scalp conductivity was set to 0.307 S/m, the relative permittivity was 149,050; the conductivity of the array coils was set to 5.998 × 107 S/m, and the relative permittivity was 1. The electrical parameters of human tissue used in this paper were provided by the online IFAC database in the United States [42]. This database provides data on the electrical properties of different tissues under the influence of electromagnetic fields from 10 Hz to 100 GHz, which is more targeted than other conductivity data [43]. (3) Add physical electromagnetic field to solve Maxwell’s equations numerically. (4) Use the transient researcher to solve the time-domain distribution of the induced electric field intensity ${\bar{E}}_{x}$ of the x-component, and calculate the low-frequency envelope-induced electric field intensity ${\bar{E}}_{nx}$ and amplitude modulation intensity $AM_{nx}$ at any position n.

### 2.3. Simulation of Multi-Target Temporal Disturbance Magnetic Stimulation in a Saline Model

We selected physiological saline to simulate human tissue fluid for simulation analysis and carried out two groups of simulation analysis. The first group of coils 1, 2, and 5 were fed with an alternating current with an amplitude of 2000 A, and the current frequency of coil 1 was set to 1.00 KHz, and the current of the other coil frequency was 1.05 KHz. The second group of coils 1, 5, and 6 was fed with an alternating current with an amplitude of 2000 A, the current frequency of coil 1 was set to 1.00 KHz, and the current frequency of the other coils was 1.05 KHz. The purpose of entering the difference frequency is to be able to modulate a low-frequency 50 Hz envelope induction electric field in the focus area. All array coils were located 5 mm from the upper surface of the cuboid. Finally, through COMSOL transient research simulation calculations, we conducted the analysis of the focus area in the time domain induction electric field situation.

### 2.4. Simulation of Multi-Target Temporal Disturbance Magnetic Stimulation in Spherical Model

We selected a five-layer sphere model to establish a simulation model of human brain multi-target temporal interference magnetic stimulation. The array coils were located 5 mm above the highest point on the sphere’s surface, and the rest of the settings were the same as those of normal saline. Then, we selected 25 mm of the lower surface of the coil as a plane parallel to the coil, which is the 20 mm depth of the sphere, and analyzed the distribution of the induced electric field of the x component.

### 2.5. Simulation of the Relationship between Temporal Disturbance Magnetic Stimulation Focal Area and Current Ratio

In the normal saline simulation model, we selected coil 1 and coil 2 to analyze the relationship between the deviation of the peak position of $AM_{nx}$ and the current ratio of the two coils, and kept the current frequency of coil 1 at 1.00 KHz and the current frequency of coil 2 at 1.05 KHz. The specific method was to keep the current frequency of coil 1 at 1.00 KHz and the current frequency of coil 2 at 1.05 KHz, fix the sum of the currents of the two coils to 4000 A, set the current ratio of Group 1 to 1:1; that is, both coils were connected to 2000 A. We set the current ratio of Group 2 to 1:5; that is, coil 1 was connected to 3333 A, and coil 2 was connected to 667 A for the simulation analysis.

## 3. Results

### 3.1. Multi-Target Temporal Interference Magnetic Stimulation in a Saline Model

The upper surface of the normal saline was 20 mm (the distance from the lower surface of the coil). The induced electric field of the x component was drawn as a plane map, and the moment was selected when the modulus was maximum. The position of the first group of simulation experiment coils is shown in Figure 4A,B. The center of the central coil was the coordinate origin, the line connecting the centers of coils 1 and 2 was the *y*-axis, the positive direction pointed to coil 2, the *x*-axis was a straight line perpendicular to the *y*-axis, and the positive direction pointed to coils 3 and 4. Figure 4C has two focal areas located between coils 1 and 2 and coils 1 and 5, respectively, which meet the needs of multi-target stimulation. In addition, there were two weaker focusing regions located on the outer edges of coil 5 and coil 2, respectively. The reason for its occurrence is related to the current flowing through the outer edge of the two coils. At the same time, this article selects the direction of the induced electric field as the x direction, so the focus area is concentrated near the *y*-axis. Despite the presence of these two weakly focused regions, coils 2 and 5 were connected with continuous high-frequency stimulation signals, so there was no cortical response in this region. We defined area 1 to be located between coils 1 and 2, and area 2 to be located between coils 1 and 5. We selected 4 positions (①②③④) in area 1 and area 2, respectively, to investigate the distribution of the induced electric field of the x component. The positions were (0, 26), (37, 26), (0, 50), (0, −26).

The green line in Figure 4D indicates point ① in area 1, which is located at the center of coils 1 and 2. At this time, ${\bar{E}}_{n1x}={\bar{E}}_{n2x}$, so the modulation effect is the best, and the $AM_{nx}$ is 4.5 V/m. (D) The middle blue line represents point ②, which is located 37 mm to the right of point ①; that is, the positive direction of the *x*-axis. Although it is far away from the center, it is at the same distance from the two coils, and still satisfies the condition of ${\bar{E}}_{n1x}={\bar{E}}_{n2x}$, but compared with point ①, the sum of $\left({\bar{E}}_{n1x}+{\bar{E}}_{n2x}\right)$ is small, so the $AM_{nx}$ will have a certain attenuation, and its value is 3.5 V/m; the black line in Figure 4E represents point ③. Located at 23 mm above ①, it is not only far away from the center of area 1, but also has an unequal distance from the two coils, resulting in an excessively large difference of $\left|{\bar{E}}_{n1x}-{\bar{E}}_{n2x}\right|$ at this point, which in turn, affects the envelope of the entire waveform; its value is 1.1 V/m. Similarly, considering the symmetry, the distribution trend of $AM_{nx}$ located in the second focal area is similar to that of area 1, as shown in the red line of point ④ in Figure 4F, and its amplitude modulation intensity $AM_{nx}$ is approximately 4.5 V/m. The arrangement of coils 1, 5, and 6 in the second group of simulations is shown in Figure 4G. Figure 4H shows the distribution of the induced electric field of the x component, and the moment with the maximum modulus was also selected for analysis. The results show that there were two high-peak focusing areas in the figure, located between coils 1 and 5 and coils 1 and 6, respectively. In addition, two low-peak focus areas were caused by the same reasons as the first group, which will be analyzed in Section 4. The reason for the two high-peak focusing areas is the same as the principle of the figure-eight coil, so this research will not go into details here, and will focus on the three points selected in Figure 4H, namely, points ⑤ (0, −26), ⑥ (−20, −16), ⑦ (−28, −46). Point ⑤ (the blue line in Figure 4I) is located at the junction of coil 1 and coil 5, and the $AM_{nx}$ was relatively high (5.6 V/m); point ⑥ is located at the junction of coil 1 and coil 6. The electric field modulus (red line in Figure 4I) shows the overall induced electric field intensity was small, but still maintained a high amplitude modulation intensity $AM_{nx}$ (5.1 V/m); point ⑦ is located at the junction of coil 5 and coil 6; theoretically, since the currents passed into the two coils were the same high frequency, the overall envelope effect was poor, and the amplitude modulation intensity was only 1.4 V/m. However, there was still a certain amplitude modulation intensity at this point, presumably due to the interference of coil 1. In summary, although there is a modulation envelope at point ⑦—the junction of coils 5 and 6—it can be ignored because the distance between coil 1 and this area is relatively small, and the interference components are small, which can be eliminated by optimizing the coil layout in the future. In addition, coils 5 and 6 are supplied with counterclockwise currents, causing the direction of the induced electric field in the junction area to be reversed, eliminating most of the induced electric field strength.

For the depth distribution of the induced electric field, the z-direction segmentation was carried out along the y-axis in Figure 4A to obtain the depth-induced electric field distribution of coils 1, 2, and 5, in physiological saline. The results are shown in Figure 5A,B. Figure 5A shows the induced electric field distribution had no temporal interference, while Figure 5B shows the $AM_{nx}$ distribution under temporal interference. By comparison, it was found that as the depth increased, the focusing area under temporal interference was significantly better than that of the group without time interference. Although the focus area was mainly concentrated at a depth of 10 mm, it may not reach the gray area. The reason is that this simulation uses a qualitative simulation analysis at a frequency of 1 KHz. To stimulate deeper areas, researchers can increase the frequency of the differential signal to the level of 10 KHz, as used in reference [14].

### 3.2. Multi-Target Temporal Interference Magnetic Stimulation in Spherical Model

The results showed that there were two focal regions in Figure 6A,B, and the positions were similar to the focal region distribution of the saline model. Similar to the physiological saline rectangular model, the spherical model still has two weak focusing regions located on the outer edges of coils 5 and 2, respectively. The reason for its occurrence is related to the current flowing through the outer edges of the two coils. Similarly, coils 2 and 5 were connected with continuous high-frequency stimulation signals, so there was no cortical response in this area. Selecting points ⑧, ⑨, and ⑩ to analyze the $AM_{nx}$, the positions were (0, 30), (35, 27), and (0, −45), respectively. The green, red, and blue lines in Figure 6C represent the induced electric field time-domain variation curves of points ⑧, ⑨, and ⑩, respectively. The results show that point ⑧ is located at the center of coils 1 and 2, so the $AM_{nx}$ was the largest; point ⑨ along the *y*-axis origin shifted 8 mm to the positive x direction, and the $AM_{nx}$ showed attenuation; point ⑩ $AM_{nx}$ was located outside the focus area, and the $AM_{nx}$ was the smallest, almost none.

The analysis of the above two models shows that the designed array coils can realize multi-target magnetic stimulation, and the AM of the low-frequency envelope-induced electric field at any target position is related to the position and the amplitude of the two high-frequency induced electric fields. The superiority of $AM_{nx}$ refers to the quality of the envelope degree of the induced electric field time-domain curve. In theory, the higher the amplitude modulation intensity, the overall amplitude of the waveform of the superimposed induced electric field is close to that of the low-frequency induced electric field, which can cause neural action. According to Equation (2), it is related to the low-frequency envelope-induced electric field amplitude modulation intensity $AM_{nx}$, which is determined by the spatial position of the target point, which is reflected in two aspects:
(a)According to Equation (2), keeping the excitation source unchanged, the upper limit of $AM_{nx}$ is related to ${\bar{E}}_{n1x}+{\bar{E}}_{n2x}$ in any point, and the maximum value of $AM_{nx}$ is ${\bar{E}}_{n1x}+{\bar{E}}_{n2x}$ (when ${\bar{E}}_{n1x}={\bar{E}}_{n2x}$).
(b)$\left|{\bar{E}}_{n1x}-{\bar{E}}_{n2x}\right|\;$ determines the superiority of $AM_{nx}$; that is, keeping the excitation source unchanged, in any point, the lower the $\left|{\bar{E}}_{n1x}-{\bar{E}}_{n2x}\right|$, the better the envelope degree of the induced electric field and the stronger the amplitude modulation intensity.


### 3.3. Relationship between the Focusing Region and Current Ratio of Temporal Interference Magnetic Stimulation

As in Section 3.1, we drew a plane parallel to the coil at 20 mm below the upper surface of the normal saline model (a distance of 25 mm from the lower surface of the coil), and drew the x component of the induced electric field. In Figure 7A, the line connecting the centers of coils 1 and 2 was first defined as the *y*-axis, and the positive direction points to coil 2. Secondly, the midpoint of the line connecting the centers of the two coils was selected as the origin of the coordinates. Finally, the *x*-axis was defined as a straight line passing through the origin and perpendicular to the *y*-axis. The direction points to coils 3 and 4. On the *x*-axis (x = 0), we selected 15 points on the left and right sides with the origin as the center, and the step length was 5 mm, which is called Array 1 (Figure 7B); similarly, we moved Array 1 positively along the *x*-axis by 5 mm, which is Array 2 (Figure 7C).

Figure 7B shows that the dot plot lines of data 1 and 2 are symmetrical, and the peak of the induced electric field $AM_{nx}$ is located at the center of the symmetry and gradually decays along the left and right, which is similar to the distribution of the induced electric field in the figure-eight focus area. In addition, the symmetry centers of $AM_{nx}$ of the two data show that there is no coincidence, and the distance was 45 mm (the peak is at the origin of the coordinates in the case of 1:1, and the peak is at the coordinate (45 mm) in the case of 1:5). The results show that changing the current ratio affects the peak position of $AM_{nx}$, and the peak position shifts to the side with smaller current, which supports Formula (2). Excluding chance, the simulation of array 2 was carried out, and the result was similar to that of array 1. As shown in Figure 7C, the peak position of $AM_{nx}$ shifted by 45 mm. However, Group 2’s overall induced electric field intensity in the two arrays in Figure 7B,C was smaller than that of the Group 1, which is described in Section 4.

To further observe the overall situation of the low-frequency-induced electric field envelope, 2 points were selected on the *y*-axis, namely a (0, 0) and b (0, 45). Figure 7D shows the envelope of point a at the ratio of 1:1 and 1:5, where the black line represents 1:1. The green line represents 1:5. The results show that the envelope of the black line was better than that of the green line. There was a certain deviation of the origin when the ratio was 1:5. Figure 7E shows the envelope of point b at the ratio of 1:1 and 1:5, where the gray line represents 1:1. The red line represents 1:5. The results show that the envelope of the red line was better than that of the gray line. This is because in the 1:5 situation, the focus area shifted.

In summary, the peak shift of $AM_{nx}$ conforms to Formula (2); that is, the overall focus area moved toward the side with a weak current. The reason is as follows, assume that the distribution of the two coils on the *y*-axis is consistent, i.e., the sum of ${\bar{E}}_{n1x}+{\bar{E}}_{n2x}$ at any point on the *y*-axis is unchanged (this is based on the symmetry of the centers of coils 1 and 2 concerning the *x*-axis), then the peak of $AM_{nx}$ appears at the maximum value of Formula (2)—when $\left|{\bar{E}}_{n1x}-{\bar{E}}_{n2x}\right|=0$. Based on the above theory, when coil 1 is fed with a current of 3333 A, which is greater than the 666 A current of coil 2, the induced electric field strength ${\bar{E}}_{n1x}$ near coil 1 is greater than the induced electric field strength ${\bar{E}}_{n2x}$ near coil 2, so the point of ${\bar{E}}_{n1x}={\bar{E}}_{n2x}$ will be biased and move to the vicinity of coil 2, the side where the current is weak.

## 4. Discussion

This paper establishes a temporal interference magnetic stimulation model with array coils to realize multi-brain region multi-targeted magnetic stimulation. Previous studies have confirmed temporal interference magnetic stimulation. It is an optimized way to balance focus and stimulation depth, especially for deep brain regions, such as the hippocampus in the memory network. In addition, by optimizing the coil position, this method can stimulate the deep target area without stimulating the superficial area, which alleviates, to a certain extent, the side effects (such as headache and dizziness) caused by the activation of the superficial cortex in the non-target area during the stimulation process. However, at present, there are many types of research on a single target in temporal interference magnetic stimulation, and it is challenging to achieve synergistic stimulation of multiple brain regions. This work aims at modeling, simulation, and analysis of multi-brain area multi-target collaborative magnetic stimulation under temporal interference, and at the same time, discusses the relationship between the focal area and current ratio of temporal interference magnetic stimulation, which is expected to contribute to the precise multi-target magnetic stimulation technology application.

Regarding the multi-target simulation problem, two models were built in this article (cuboid saline and five-layer ball model) for multi-target correlation analysis. First, in normal saline, two sets of simulations were conducted. The purpose was to achieve multi-targeted focus areas. The selection principle of these two groups of simulation coils was to fully demonstrate the purpose of simulation in this section and to establish a reasonable simulation process. In this regard, the first group, namely, conduction coils 1, 2, and 5, aimed to investigate whether a multi-target focal area could be formed under temporal interference. However, this result can show that this group of similarly arranged coils in the same column can produce multiple foci, and there will be no obvious interference; in the second group, the purpose is to check whether there will be unnecessary interference between adjacent coils, so coils 1, 5, and 6 were selected for analysis, and the results show that although the distance between coils 5 and 6 is relatively close, there was no obvious focus area. The reasons are as follows. The first reason is that the current directions of coils 5 and 6 are the same, resulting in a large attenuation of the induced electric field superposition in the focus area; the second reason is that in coils 5 and 6, the same frequency current was passed through so that it would not produce envelope superposition in the intersecting area. To sum up, the first group explains that the coils in the same column can generate multiple target focus areas, and the second group explains whether there is interference between adjacent coils with the same frequency. In addition, after careful consideration in this paper, it is believed that the cross structure in Figure 8A (that is, the cross structure centered on coil 1) and the adjacent coil (same frequency) in Figure 8B are far away from each other and have the same frequency, so the focus area occurs around coil 1. Hence, the author of this paper believes that the symmetrical arrangement and expansion of the coils can be carried out on the first and second sets of simulations to realize the role of multiple targets.

In the simulation results of the multi-target temporal interference magnetic stimulation saline and the sphere model, there are two high-peak and two low-peak focus areas in Figure 4C,H, which were located between coil 1 and coil 2, as well as coil 1 and coil 5, respectively. The reason is the same as the figure-eight coil. The focus of the two lower peaks comes from the edge of the circular coil. Since the x component of the induced electric field is extracted in this paper, the x component of the induced electric field near the *y*-axis accounts for the dominant force, so the focus area is mainly concentrated around the *y*-axis of the three circular coils. This paper analyzes the modulation of the two low-peak focus areas, selects point ④’ (0, −90) located at the edge of coil 5 in Figure 9A as the analysis point, compares point ① in 3.1, and calculates the x-component of the two points through simulation of the time-domain change curve of the induced electric field. The green line in Figure 9B represents point ①, and the blue line represents point ④’. The results show that the envelope shape of the curve at point ④’ is worse than point ①, and the peak value of the induced electric field at the two points differs by 2.8 V/m (①: 6.3 V/m, ④’: 3.5 V/m); the $AM_{nx}$ differs by 3 V/m (①: 4.1 V/m, ④’: 1.1 V/m). The data show that although there are two low-peak focal regions, their amplitude modulation intensity is greatly reduced compared with the high-peak focal region. It can be found in Figure 4H,I that the focus area is still in the coil group with different frequencies (coils 1, 5 and coils 1, 6) while maintaining a high amplitude modulation intensity. However, the $AM_{nx}$ of the spot found between coils 5 and 6 was weaker, indicating that the difference frequency induced electric field is a sufficient condition for the modulation envelope. To sum up, the proposed structure can realize multiple focal areas in the low-frequency envelope of the brain region through the first group. At the same time, the second group shows that adjacent coils will not add extra focal areas. In the next work, we change the size of the coil and expand the number of coils to *n* in a clockwise direction to achieve temporal interference magnetic stimulation for more targets. In addition, the focal zone fine-tuning technique mentioned in Section 3.3 optimizes the coil current ratio, and the position of the focal zone can be adjusted precisely.

The simulation model was built with physiological saline to analyze the relationship between the change of the focus area of temporal interference magnetic stimulation and the current ratio. Figure 7B shows that when the ratio changed from 1:1 to 1:5, the focus area shifted from the center of the coil, i.e., the origin to 45 mm, and there was still symmetry, indicating that the current adjustment ratio can be shifted from the peak position of $AM_{nx}$, thereby changing the position of the overall focus area. The offset of the region is consistent with the magnitude mentioned in the literature [12], both at the centimeter level. From Figure 7B,C, this study also found that the overall induced electric field modulus of Group 2 (1:5) in arrays 1 and 2 was smaller than that of Group 1 (1:1) because the simulation in Section 3.3 was based on the sum of the two coils maintaining. Therefore, when the ratio is changed, the focus area tends to move towards the direction of the low induced electric field intensity, resulting in the overall intensity being smaller than that of Group 1. In summary, this paper simulates the relationship between the movement of the focus area of temporal interference magnetic stimulation and the current ratio and selects two coils with different frequencies for verification and analysis. The author believes that the key to fine-tuning the focus lies in the ratio of the induced electric field of different frequencies in the focus area rather than the number of activated coils, so this paper does not conduct simulation with multiple coils (the combination of multiple coils at two frequencies is essentially a superposition of fields).

For future work, the author believes that the precise adjustment of the focus area position can be achieved by optimizing the coil structure through algorithms, designing the coil space topology, and adjusting the ratio of different frequency currents. In addition, by collecting multi-channel EEG signals and MRI images of patients before and after stimulation, and based on the characteristics of behavioral scale-EEG-MRI image multimodal data fusion, a TMS electromagnetic closed-loop control scheme is designed.

## 5. Conclusions

In this study, we propose a multi-target temporal interference magnetic stimulation system with array coils, which integrates multi-brain region co-stimulation technology with temporal interference, aiming to stimulate multiple targets in the deep brain. We validated this technique in saline and human brain sphere models. In addition, we adopted the method of shifting the focus area without moving the coil by changing the current ratio between coils with different frequencies and verified the feasibility of this method through simulation. These results indicate that the multi-target temporal interference magnetic stimulation with the array coils can non-invasively stimulate multiple network nodes in the brain simultaneously, and precise positioning can be achieved by controlling the conduction of different coils and changing the proportion of the current in the coils.

## Figures and Tables

**Figure 1 bioengineering-10-00568-f001:**
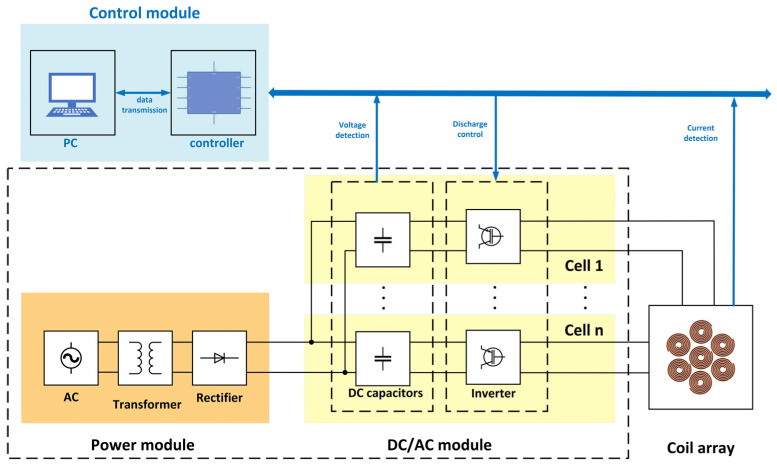
Multi-target temporal interferometry system with array coils.

**Figure 2 bioengineering-10-00568-f002:**
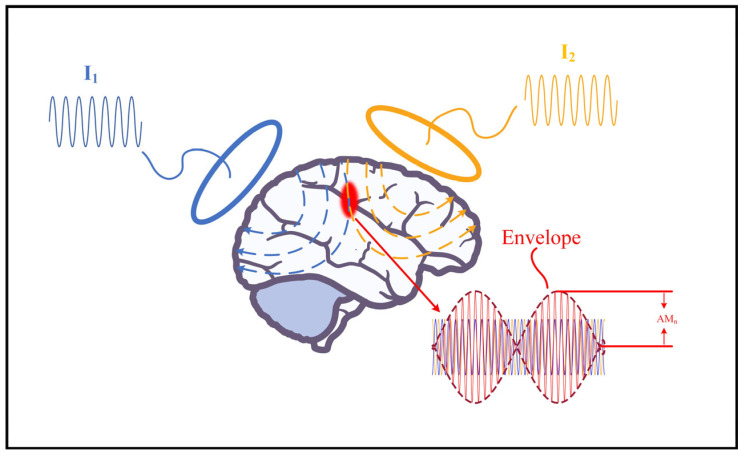
The temporal interference magnetic stimulation model. The light red line represents the low-frequency envelope modulation waveform, *AM_n_* represents the amplitude modulation intensity, the blue line represents the electric field waveform with a frequency of 1.05 KHz, the orange line represents the electric field waveform with a frequency of 1.00 KHz, and the red line represents the electric field waveform with a frequency of 50 Hz.

**Figure 3 bioengineering-10-00568-f003:**
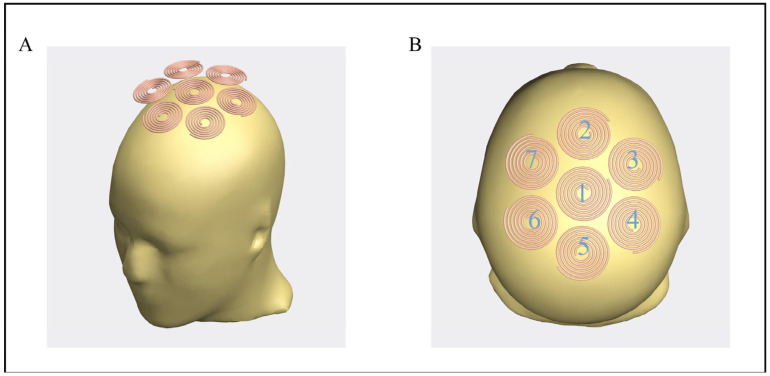
The temporal interference magnetic stimulation model with array coils. (**A**) Schematic diagram of the structure of the coil arrangement. (**B**) Top view of the coil arrangement, the numbering rules for the seven coils in the figure are as follows: the middle coil is numbered as 1, and the remaining coils are arranged clockwise from above the coil numbered 1 as 2, 3, 4, 5, 6, and 7.

**Figure 4 bioengineering-10-00568-f004:**
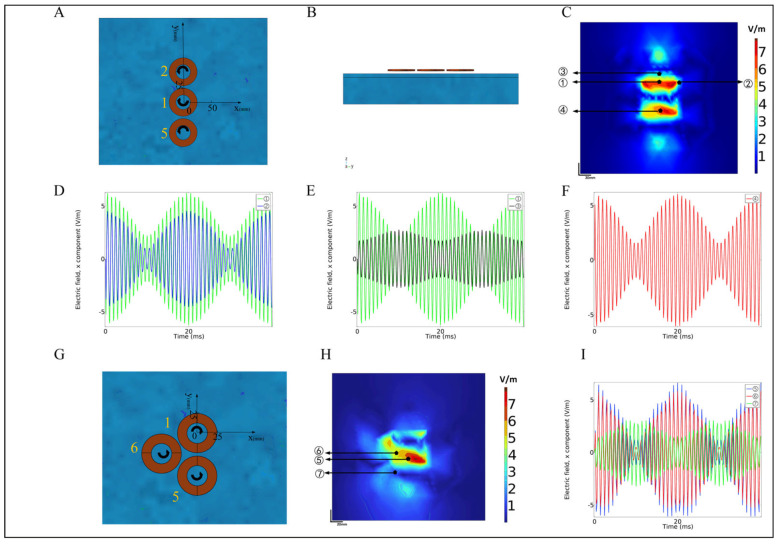
(**A**) Schematic diagram of the spatial positions of coils 1, 2, and 5 in a saline model, with arrows indicating the direction of coil current. (**B**) Schematic diagram of the arrangement side of coils 1, 2, and 5 in the saline water model. (**C**) Simulation results of coils 1, 2, and 5 focus areas. (**D**) Point ①, ② time-domain distribution of induced electric field. (**E**) Time-domain distribution of induced electric field at points ① and ③. (**F**) Time-domain distribution of induced electric field at point ④. (**G**) Schematic diagram of the spatial positions of coils 1, 5, and 6 in a saline model, with arrows indicating the direction of coil current. (**H**) Simulation results of coils 1, 5, and 6 focus areas. (**I**) Time-domain distribution of induced electric field at points ⑤, ⑥, ⑦.

**Figure 5 bioengineering-10-00568-f005:**
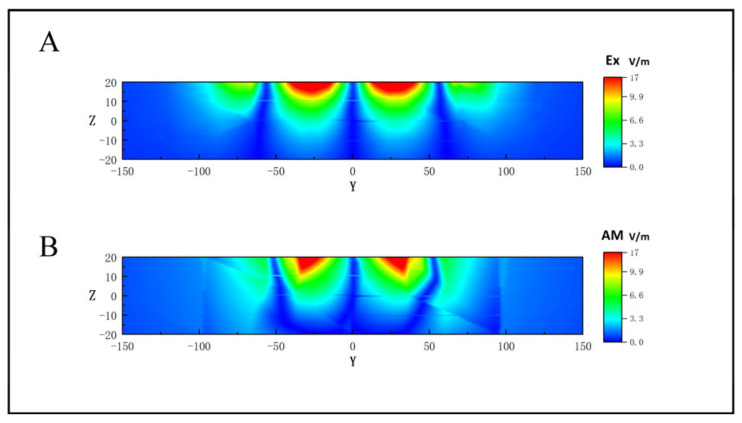
Depth distribution map of induced electric field. (**A**) Distribution of deep induction electric field no temporal interference. (**B**) Distribution of $AM_{nx}$ temporal interference.

**Figure 6 bioengineering-10-00568-f006:**
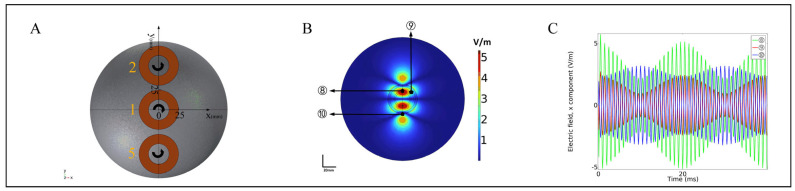
Sphere model simulation results. (**A**) Schematic diagram of the arrangement of coils 1, 2, 5 under the sphere head model. (**B**) Simulation of two focus areas in the sphere head model. (**C**) Temporal distribution of induced electric field at points ⑧, ⑨, ⑩.

**Figure 7 bioengineering-10-00568-f007:**
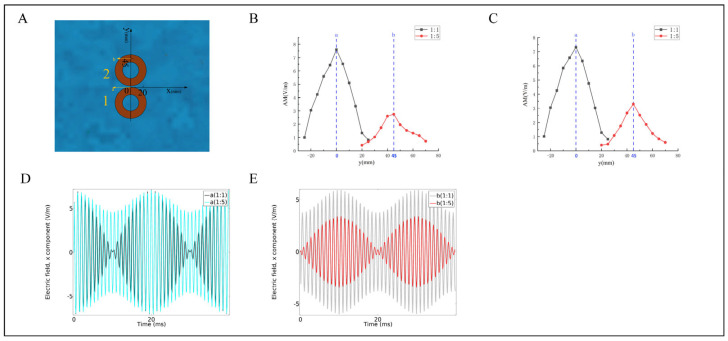
The simulation results of the movement of the focus area. (**A**) Schematic diagram of coil placement. (**B**) Array 1: the dotted line diagram at x = 0, represented by a black line marked with a box for 1:1, and a red line marked with a circle for 1:5. (**C**) Array 2: the dotted line diagram at x = 5, represented by a black line marked with a box for 1:1, and a red line marked with a circle for 1:5. (**D**) The induced electric field distribution at point a at the ratio of 1:1 and 1:5, when x = 0. (**E**) Induced electric field distribution at point b at a ratio of 1:5, when x = 0.

**Figure 8 bioengineering-10-00568-f008:**
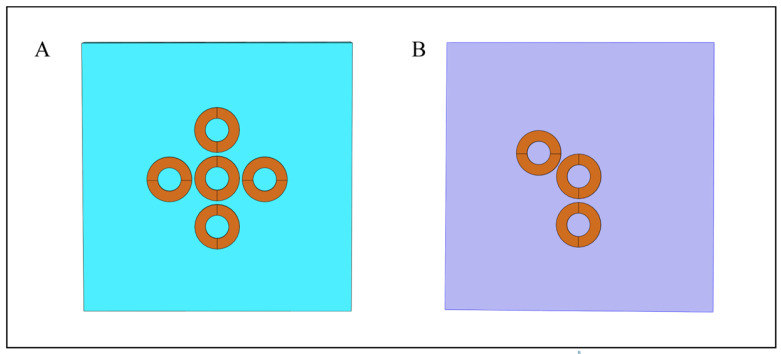
Schematic diagram of array coil unit arrangement. (**A**) The cross structure. (**B**) Adjacent coil structures with long distances.

**Figure 9 bioengineering-10-00568-f009:**
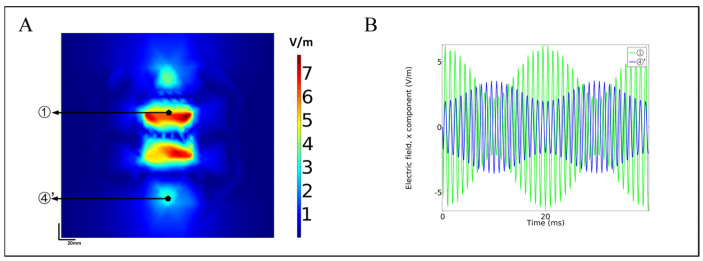
(**A**) ①, ④’ focus area, (**B**) ①, ④’ time-domain comparison of induced electric field.

## Data Availability

Not applicable.

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
