# Peer review of "Optimal Design of Array Coils for Multi-Target Adjustable Electromagnetic Brain Stimulation System"

_bioengineering, 2023, doi:10.3390/bioengineering10050568_

Round 1

Reviewer 1 Report

The authors proposed a multi-target temporal interference magnetic stimulation system with array coils which can simultaneously stimulate multiple targets in deep brain regions. I believe that this kind of technology will be of great help to patients if it can be applied to the clinic. However, there are several problems with the manuscript that need to be solved.

1.      The language of this article needs polishing.

2.      The authors say that the coil radius is 25mm, the wire diameter is 2mm and the number of turns is 6. So are these parameters the most appropriate for coil performance and experimental results. What is the effect of changing these parameters?

3.      The references could have been a little richer, making the language of the manuscript more convincing.

Author Response

Dear  Reviewer:

Thank you for reviewing our manuscript

entitled “Optimal Design of Array Coils for Multi-target Adjustable Electromagnetic Brain Stimulation System” (ID: bioengineering-2314035). Those comments are all valuable and very helpful for revising and improving our paper, as well as the important guiding significance to our researches. We have studied comments carefully and have made correction which we hope meet with approval. The details of the modification are in the attachment.

                                  Author:Dr. Tingyu Wang、Mr. Lele Yan、Dr. Xinsheng Yang、Prof. Duyan Geng、Prof. Dr. Guizhi Xu *、Prof. Alan Wang *

  1. The language of this article needs polishing.

Thank you for your lack of wording and expression in the article. We reviewed the full text and revised a series of inappropriate sentences.

  • Original [47-]:

Among them, the principle of transcranial magnetic stimulation technology is to place the pulsed magnetic field generated by the energized coil above the scalp, through the skull to reach the targeted brain area and generate an induced electric field, which will depolarize neurons and generate action potentials to regulate the neuronal activity, and then affect the cognitive function of the brain [4-6].

Modified version [45-]:

Among them, the principle of transcranial magnetic stimulation technology is to place the pulsed magnetic field generated by the energized coil above the scalp, through the skull to reach the targeted brain area and generate an induced electric field, which will make neurons depolarization and generate action potential, thus affecting the metabolism and electrical activity of nerves in the brain [4-7].

  • Original [169-]:

There are two main tasks for simulating multi-target temporal interference magnetic stimulation: (1) Select coils 1, 2, and 5 to feed in differential frequency alternating current, and perform simulations in human tissue fluid and human brain sphere to realize multi-target array coils temporal interference: choose coils 1, 5, and 6 to pass in differential frequency alternating current, and simulate in the human tissue fluid model to analyze the relationship between the focus area and the arrangement of the coils. (2) The total current of fixed coils 1 and 2 is 4000A, the current frequency of coil 1 is kept at 1.05KHz, and the current frequency of coil 2 is 1.00KHz. By adjusting the current ratio of the two coils, observe the change in the peak position of , and analyze the relationship between the movement of the focus area and the current ratio of coils 1 and 2.

Modified version [215-]:

There are two main tasks for simulating multi-target temporal interference magnetic stimulation: (1) Implementation of multi-target array coil temporal interference: This paper conducted temporal interference simulations of two sets of physiological saline models for multi-target array coils, with the first set being conducting coils 1, 2, and 5; The second group is the conduction coils 1, 5, and 6. In addition, in the ball model, the distribution of the focusing area was analyzed when coils 1, 2, and 5 were turned on. (2) Analyze the relationship between the current ratio of two different frequency coils and the movement of the focusing area: apply a 1KHz AC signal to coil 1 and a 1.05KHz AC signal to coil 2, adjust their current ratio, and observe the changes in the focusing area.

  • Original [56-]:

Temporal interference transcranial magnetic stimulation is a simulation method similar to temporal interference transcranial electrical stimulation [8-9]. The inner stimulation target superimposes a low-frequency envelope induction electric field, and the area where the non-target target is located is a high-frequency signal. Due to the low-pass characteristic of the neuron membrane, the neurons in the area where the envelope is not generated will not respond, thereby activating the deep part of the brain, the role of local area neurons.

Modified version [58-]:

According to references [9-15], both temporal interference transcranial magnetic stimulation and temporal interference transcranial electrical stimulation are deep brain stimulation schemes based on the principle of temporal interference. Adjusting the spatial position of the coil can generate a low-frequency envelope-induced superimposed electric field in the target area, making neurons follow this low-frequency envelope oscillation to generate electrical activity; In nontarget areas, superimposed induction electric fields are dominated by high-frequency induction electric fields, and combined with the extensively accepted low-pass filtering characteristic mechanism of neuronal membranes, the neural electrical activity in this area will not be able to follow very-high-frequency oscillating electric fields[13].

  • Original [290-]:

The analysis of the above two models shows that the designed array coils can realize multi-target magnetic stimulation, and the  of the low-frequency envelope induced electric field at any target position is related to the position and the amplitude of the two high-frequency induced electric fields. value dependent.

Modified version [394-]:

The analysis of the above two models shows that the designed array coils can realize multi-target magnetic stimulation, and the  of the low-frequency envelope induced electric field at any target position is related to the position and the amplitude of the two high-frequency induced electric fields.

  • Original [288-]:

The analysis of the above two models shows that the designed array coils can realize multi-target magnetic stimulation, and the  of the low-frequency envelope induced electric field at any target position is related to the position and the amplitude of the two high-frequency induced electric fields. value dependent. Definition The superiority of  refers to the quality of the envelope degree of the induced electric field time-domain curve. According to the formula (1), it is related to the low-frequency envelope induced electric field modulation intensity, which is determined by the spatial position of the target point, which is reflected in Two aspects:

(a)  The size of determines the upper limit of .

(b) determines the superiority of , that is, under certain circumstances of , the smaller , the better the envelope degree of the induced electric field and the stronger the modulation intensity high, and vice versa.

Modified version [394-]:

The analysis of the above two models shows that the designed array coils can realize multi-target magnetic stimulation, and the  of the low-frequency envelope induced electric field at any target position is related to the position and the amplitude of the two high-frequency induced electric fields. Definition the superiority of  refers to the quality of the envelope degree of the induced electric field time-domain curve. In theory, the higher the amplitude modulation intensity, the overall amplitude of the waveform of the superimposed induced electric field is close to that of the low-frequency induced electric field, which can cause neural action. According to the equation (2), it is related to the low-frequency envelope induced electric field amplitude modulation intensity  , which is determined by the spatial position of the target point, which is reflected in two aspects:

  • According to equation 2, keep the excitation source unchanged, the upper limit of is related to  in any point, and the maximum value of  is .(when  ).

determines the superiority of , that is, keep the excitation source unchanged, in any point, the lower the , the better the envelope degree of the induced electric field and the stronger the amplitude modulation intensity high.

2.The authors say that the coil radius is 25mm, the wire diameter is 2mm and the number of turns is 6. So are these parameters the most appropriate for coil performance and experimental results. What is the effect of changing these parameters?

Thanks for your question about the effect of coil size on the results:

The size of the coil determines the spatial distribution of the induced electric field, that is, determines the spatial characteristics of the magnetic stimulation. Specifically, as the size of the coil increases, the intensity of the electric field induced in the cortex becomes higher, and the depth of the electric field attenuates weaker; , the increase in size will affect its focusing effect, and the focusing performance of small-sized coils is better than that of large-sized coils. So, the maximum limit of the magnetic stimulation coil when weighing depth-focus-strength. Finally, we describe the relevant content in the text:

Original [131-]:

All the coils are circular coils of the same size, and the arrangement is shown in Figures 3A and 3B. The center of the circular coil numbered 1 coincides with the center of the array, which is called the center coil; then the remaining numbered coils (from small to large) are arranged clockwise around the center coil. The coil radius is 25mm, the wire diameter is 2mm, the number of turns is 6, and the material is metallic copper with good conductivity.

Modified version [163-]:

All coils were circular coils of the same size, arranged as shown in Figures 3A and 3B. The center of the circular coil numbered 1 coincides with the center of the array, which is called the central coil; the remaining numbered coils (from small to large) are arranged clockwise around the central coil. The selection of radius has two factors: firstly, considering that the diameter of the five-layer sphere model is 184mm, the head circumference of adults is between 540-580mm, and there are at least three circles in the same column; Secondly, the coil radius is approximately positively correlated with the stimulus intensity. Specifically, as the coil radius increases, the cortical stimulus intensity gradually increases. So the coil unit is selected with an outer radius of 25mm [29]. The wire diameter is 2mm, the number of turns is 6, and the material is copper with good conductivity. The shell is made of an insulating epoxy resin board, which is characterized by good mechanical properties under medium temperature conditions, stable electrical properties under high humidity conditions, good heat resistance and moisture resistance, and can protect personnel safety .

  1. The references could have been a little richer, making the language of the manuscript more convincing.

Thank you for your suggestion about too few references and insufficient convincing power. We have made relevant supplements. Thank you again for your opinions and wish you a happy life.

References

[1] Vissani, Matteo, Ioannis U Isaias, and Alberto Mazzoni. "Deep Brain Stimulation: A Review of the Open Neural Engineering Challenges." Journal of neural engineering 17, no. 5 (2020): 051002.DOI:10.1088/1741-2552/abb581

[2] Song, Xizi, Xue Zhao, Xiaohong Li, Shuang Liu, and Dong Ming. "Multi-Channel Transcranial Temporally Interfering Stimulation (Ttis): Application to Living Mice Brain." Journal of neural engineering 18, no. 3 (2021): 036003.DOI:10.1088/1741-2552/abd2c9

[3] Zhu Haoran, Huai Ruituo, Zhang Pingqiu, and Wang Hui. "Research progress of noninvasive neuromodulation methods based on temporal interference." Chinese Journal of Medical Physics 40, no. 1 (2023): 6.DOI:10.3969/j.issn.1005-202X.2023.01.018

[4] Barker, Anthony T, Reza Jalinous, and Ian L Freeston. "Noninvasive Magnetic Stimulation of Human Motor Cortex." The Lancet 325, no. 8437 (1985): 1106-07.DOI:10.1016/S0140-6736(85)92413-4

[5] Li Jiangtao, Zheng Minjun, and Cao Hui. "Research Progress of Transcranial Magnetic Stimulation Technology." High Voltage Technology 42, no. 4 (2016): 11.DOI:10.13336/j.1003-6520.hve.201604005037

[6] Xia Siping, Xu Yajie, Yu Yingcong, Gu Weiguo, Ma Changyu, and Yang Xiaodong. "Research Progress in Transcranial Magnetic Stimulation Electric Field Analysis." Chinese Journal of Biomedical Engineering 39, no. 6 (2020): 9.DOI:10.3969/j.issn.0258-8021.2020.06.010

[7] Gutierrez M I, Poblete-Naredo I, Mercado-Gutierrez J A, et al. Devices and Technology in Transcranial Magnetic Stimulation: A Systematic Review[J]. Brain Sciences, 2022, 12(9): 1218.

Xiong Hui, Jing Zhao, and Liu Jinzhen. "Research progress of transcranial magnetic stimulation system." Aerospace Medicine and Medical Engineering 33, no. 6 (2020): 9.DOI:10.16289/j.cnki.1002-0837.2020.06.012

[9] Grossman, Nir, David Bono, Nina Dedic, Suhasa B Kodandaramaiah, Andrii Rudenko, Ho-Jun Suk, Antonino M Cassara, Esra Neufeld, Niels Kuster, and Li-Huei Tsai. "Noninvasive Deep Brain Stimulation Via Temporally Interfering Electric Fields." Cell 169, no. 6 (2017): 1029-41. e16.https://doi.org/10.1016/j.cell.2017.05.024.

[10] Guo, Wanting, Yuchen He, Wenquan Zhang, Yiwei Sun, Junling Wang, Shuang Liu, and Dong Ming. "A Novel Noninvasive Brain Stimulation Technique: "Temporally Interfering Electrical Stimulation"." Frontiers in Neuroscience 17 (2023).Doi:10.3389/fnins.2023.1092539.

[11] Cao J, Grover P. Stimulus: Noninvasive dynamic patterns of neurostimulation using spatio-temporal interference[J]. IEEE Transactions on Biomedical Engineering, 2019, 67(3): 726-737.

[12] Sorkhabi, Majid Memarian, Karen Wendt, and Timothy Denison. "Temporally Interfering Tms: Focal and Dynamic Stimulation Location." Paper presented at the 2020 42nd Annual International Conference of the IEEE Engineering in Medicine & Biology Society (EMBC) 2020.DOI:10.1109/EMBC44109.2020.9176249

[13] Xin, Zonghao, Akihiro Kuwahata, Shuang Liu, and Masaki Sekino. "Magnetically Induced Temporal Interference for Focal and Deep-Brain Stimulation." Frontiers in Human Neuroscience 15 (2021): 693207.https://doi.org/10.3389/fnhum.2021.693207

[14] Khalifa, Adam, Seyed Mahdi Abrishami, Mohsen Zaeimbashi, Alexander D Tang, Brian Coughlin, Jennifer Rodger, Nian X Sun, and Sydney S Cash. "Magnetic Temporal Interference for Noninvasive and Focal Brain Stimulation." Journal of neural engineering 20, no. 1 (2023): 016002.DOI:10.1088/1741-2552/acb015

[15] Greenberg M E, Hermanowski A L, Ziff E B. Effect of protein synthesis inhibitors on growth factor activation of c-fos, c-myc, and actin gene transcription[J]. Molecular and cellular biology, 1986, 6(4): 1050-1057.

[16] Goncalves S B, Palha J M, Fernandes H C, et al. LED optrode with integrated temperature sensing for optogenetics[J]. Micromachines, 2018, 9(9): 473.

[17] Nordi T M, Gounella R H, Luppe M, et al. Low-Noise Amplifier for Deep-Brain Stimulation (DBS)[J]. Electronics, 2022, 11(6): 939.

[18] Li Jiangtao, Cao Hui, Zheng Minjun, and Zhao Zheng. "Drive and Control of Multi-Channel Transcranial Magnetic Stimulation Coil Array." Journal of Electrotechnical Society 32, no. 22 (2017): 8.DOI:10.19595/j.cnki.1000-6753.tces.170142

[19] Yang, Shuo, Guizhi Xu, Lei Wang, Yaohua Geng, Hongli Yu, and Qingxin Yang. "Circular Coil Array Model for Transcranial Magnetic Stimulation." IEEE Transactions on Applied Superconductivity 20, no. 3 (2010): 829-33.DOI: 10.1109/TASC.2010.2040379

[20] Ho, Siu Lau, Guizhi Xu, WN Fu, Qingxin Yang, Huijuan Hou, and Weili Yan. "Optimization of Array Magnetic Coil Design for Functional Magnetic Stimulation Based on Improved Genetic Algorithm." IEEE Transactions on Magnetics 45, no. 10 (2009): 4849-52.DOI:10.1109/TMAG.2009.2025892

[21] Xiong Hui, Gao Yijuan, Liu Jinzhen Research on coil array design method based on transcranial magnetic stimulation [J] Aerospace Medicine and Medical Engineering, 2018, 31 (5): 545-550

[22] Xiong Hui, Qiu Bowen, Liu Jinzhen Simulation study of multi-channel transcranial magnetic stimulation cap coil unit based on MRI data [J] Aerospace Medicine and Medical Engineering, 2020, 33 (3): 246-251

[23] Liu Fucai, Jin Shuhui, Zhao Xiaojuan Comparison of LC series resonance and LCC series parallel resonance in high-voltage pulse capacitor charging power supply [J] High Voltage Technology, 2012, 38 (12): 3347-3356

[24] Peterchev A V, Jalinous R, Lisanby S H. A transcranial magnetic stimulator inducing near-rectangular pulses with controllable pulse width (cTMS)[J]. IEEE Transactions on Biomedical Engineering, 2007, 55(1): 257-266.

[25] Peterchev A V, Murphy D L, Lisanby S H. Repetitive transcranial magnetic stimulator with controllable pulse parameters[J]. Journal of neural engineering, 2011, 8(3): 036016.

[26] Sorkhabi M M, Benjaber M, Wendt K, et al. Programmable transcranial magnetic stimulation: a modulation approach for the generation of controllable magnetic stimuli[J]. IEEE Transactions on Biomedical Engineering, 2020, 68(6): 1847-1858.

[27] Gattinger N, Moßnang G, Gleich B. flexTMS—a novel repetitive transcranial magnetic stimulation device with freely programmable stimulus currents[J]. IEEE transactions on biomedical engineering, 2012, 59(7): 1962-1970.

[28] Sorkhabi M M, Denison T. A neurostimulator system for real, sham, and multi-target transcranial magnetic stimulation[J]. Journal of Neural Engineering, 2022, 19(2): 026035.

[29] Chen Hailei Research and design of a multi parameter controllable transcranial magnetic stimulation coil [D] Chongqing University of Posts and Telecommunications, 2019

[30] Ueno, S, T Tashiro, and K Harada. "Localized Stimulation of Neural Tissues in the Brain by Means of a Paired Configuration of Time‐Varying Magnetic Fields." Journal of Applied Physics 64, no. 10 (1988): 5862-64.https://doi.org/10.1063/1.342181

[31] Deng, Zhi-De, Sarah H Lisanby, and Angel V Peterchev. "Electric Field Depth–Focality Tradeoff in Transcranial Magnetic Stimulation: Simulation Comparison of 50 Coil Designs." Brain stimulation 6, no. 1 (2013): 1-13.https://doi.org/10.1016/j.brs.2012.02.005

[32] Guadagnin V, Parazzini M, Fiocchi S, et al. Deep transcranial magnetic stimulation: modeling of different coil configurations[J]. IEEE Transactions on Biomedical Engineering, 2015, 63(7): 1543-1550.

[33] Deng Z D, Lisanby S H, Peterchev A V. Coil design considerations for deep transcranial magnetic stimulation[J]. Clinical Neurophysiology, 2014, 125(6): 1202-1212.

[34] Krieg T D, Salinas F S, Narayana S, et al. Computational and experimental analysis of TMS-induced electric field vectors critical to neuronal activation[J]. Journal of neural engineering, 2015, 12(4): 046014.

[35] Roth, Bradley J, and Peter J Basser. "A Model of the Stimulation of a Nerve Fiber by Electromagnetic Induction." IEEE Transactions on Biomedical Engineering 37, no. 6 (1990): 588-97.DOI:10.1109/10.55662

[36] Chunye, R., et al. (1995). "A novel electric design for electromagnetic stimulation-the Slinky coil." IEEE Transactions on Biomedical Engineering 42(9): 918-925.

[37] Afuwape, O. F., et al. (2021). "Measurement and Modeling of the Effects of Transcranial Magnetic Stimulation on the Brain." IEEE Transactions on Magnetics 57(2): 1-5.

[38] de Lara, L. I. N., et al. (2021). "A 3-axis coil design for multichannel TMS arrays." NeuroImage 224: 117355.

[39] Zhao Chen (2011) Simulation and coil optimization of intracranial induced electric field distribution and energy distribution under magnetic stimulation, Beijing Union Medical College.

[40] Hese P V ,  Vanrumste B ,  Boon P , et al. Dipole Localization Errors due to not Incorporating Compartments with Anisotropic Conductivities: Simulation Study in a Spherical Head Model[J]. International Journal of Biolelectromagnetism, 2005, 7(1):134-137.

[41] Zhu, X., et al. (2019). "Multi-Point Temporal Interference Stimulation by Using Each Electrode to Carry Different Frequency Currents." IEEE Access 7: 168839-168848.

[42] D.Andreuccetti, R.Fossi and C.Petrucci: An Internet resource for the calculation of the dielectric properties of body tissues in the frequency range 10 Hz - 100 GHz. IFAC-CNR, Florence (Italy), 1997. Based on data published by C.Gabriel et al. in 1996. [Online]. Available:http://niremf.ifac.cnr.it/tissprop/

[43] Chen Zhao (2017). Transcranial magnetic stimulation simulation analysis based on real cranial structure modeling, Peking Union Medical College; Chinese Academy of Medical Sciences; Tsinghua University Health Science Center; Peking Union Medical College, Chinese Academy of Medical Sciences.

Reviewer 2 Report

The study by Wang et al. provides an interesting design for multi-coil TMS stimulation. The aim is to perform multi-target and deep TMS using interference, which is an ambitious goal of evident importance.

Nevertheless, the aim si scarcely achieved by the authors and the manuscript should require substantial revision in terms of analyses and writing.

At first the writing is very poor and several errors are met. In addition description of important aspects is often badly phrased if not impossible to understand:

- l47 "the cognitive function" is a very basic way of speaking about brain and mind as in many other instances

- l56-60 unintelligible

- 74-75 unintelligible cFOS description

- 169-177 tasks are explained in an unclear way, also using a bad writing form as occurs often in the text, please avoid using lists of "imperative" sentences.

- 185 what is the aim of this cuboid? What does it model? Why these dimensions?

- 187 the sphere: what is the depth of each layer? Is it based on a realistic model of the brain? of course not. Why is it not possible to use a realistic brain model which can be obtained by other sources and open source software?

- 290 orphan sentence

- 292-298 not intelligible

- 320 should it be 1:1?

- 319 why using experimental and control? These are not experiments or experimental groups rather different simulating conditions

- 410 orphan sentence

Figure 4C and 5: details are not appreciable, especially the location and text of point in circles. Colormaps miss the units.

What is the depth of these points? A 3D representation of the simulation map should be provided to locate the real depth of fields of significant intensity.

Figure 4F wrong points in legends (1), figure 4E in legend states two times the same point.

In Figure 4A and G coil number should be indicated

Figure 4C and 5B: line 274 states that only two focal spots are produced, but 4 are visible. Two additional focus points are presente (red in the map) at the extreme sides and should be analyzed and discussed.

Figure 5D a(1:3) plot is not coherent with AM plot in Figure 5B

442-447, please rephrase: these are too ambitious aims and present form is used which seems to refer to work in progress.

Title: nerve is misleading (the aim is brain stimulation here). Intelligent optimization is misleading since there is no optimization in this manuscript and no use of AI. Also remove any reference in the text to these aspects (intelligent is often used). 

Limitations must be discussed including the positioning of the device on the skull, the possibility to scale the stimulation locations to any skull or brain size etc.

More importantly, a major limitation of this study refers to:

1) lack of high depth stimulation: the simulations and analyses are performed at 5-10 mm from surface, which is very limited and does not makes any sense in the context of deep-TMS applications, such as sub-cortical nuclei. In addition, no 3D visualization is provided to really understand the position of the focal spots in depth.

2) simultaneous AM and multi-site stimulation is not shown, rather these are only separately discussed, contrary to the main scope of the study. In addition, only specific cases (a few) are discussed, and very few images of simulations are provided, so that that it is difficult to generalize and to appreciate the result. For instance, no 3D colormap representation of AM effect is provided while this would be the most crucial aspect of this study.

Author Response

Dear  Reviewer :

Thanks very much for your kind letter about the review of our paper titled “Optimal Design of Array Coils for Multi-target Adjustable Electromagnetic Brain Stimulation System” (ID: bioengineering-2314035). for publication in “bioengineering”. We have finished the proof reading and checking carefully, and some corrections about the answers to the queries are provided below.The details of the modification are in the attachment.

                                                     Author:Dr. Tingyu Wang、Mr. Lele Yan、Dr. Xinsheng Yang、Prof. Duyan Geng、Prof. Dr. Guizhi Xu *、Prof. Alan Wang *

  1. 47 "the cognitive function" is a very basic way of speaking about brain and mind as in many other instances

We have made the following modifications regarding the issue of inappropriate vocabulary for the "the cognitive function".

Original [47-]:

Among them, the principle of transcranial magnetic stimulation technology is to place the pulsed magnetic field generated by the energized coil above the scalp, through the skull to reach the targeted brain area and generate an induced electric field, which will depolarize neurons and generate action potentials to regulate the neuronal activity, and then affect the cognitive function of the brain [4-6].

Modified version [45-]:

Among them, the principle of transcranial magnetic stimulation technology is to place the pulsed magnetic field generated by the energized coil above the scalp, through the skull to reach the targeted brain area and generate an induced electric field, which will make neurons depolarization and generate action potential, thus affecting the metabolism and electrical activity of nerves in the brain [4-7].

  1. 56-60 unintelligible

We are very sorry that due to our negligence in writing, the reader was unable to understand the paragraph. Therefore, we have provided a detailed explanation. Thank you again for your feedback. Here is our detailed explanation:

Original [56-]:

  • Temporal interference transcranial magnetic stimulation is a simulation method similar to temporal interference transcranial electrical stimulation [8-9].
  • The inner stimulation target superimposes a low-frequency envelope induction electric field, and the area where the non-target target is located is a high-frequency signal.
  • Due to the low-pass characteristic of the neuron membrane, the neurons in the area where the envelope is not generated will not respond, thereby activating the deep part of the brain, the role of local area neurons.

The detailed description we provide:

  • According to references [8-9], Both temporal interference transcranial magnetic stimulation and temporal interference transcranial electrical stimulation are deep brain stimulation schemes based on the principle of temporal interference.
  • As shown in Figure 1 and equation 1, the red area (target area) is composed of two closely or completely identical differential induction electric fields, resulting in a higher amplitude modulation intensity of the low-frequency envelope induction electric field in this area, which is close to the low-frequency induction electric field; However, in other areas (nontarget areas), such as the area below the blue coil, the amplitude modulation intensity is poor due to the superposition of two different frequency induction electric fields with significant differences, which is close to the high-frequency induction electric field.

,                       (1)

Figure 1. The temporal interference magnetic stimulation model (Represents the low-frequency envelope modulation waveform, represents the modulation intensity, the blue line represents the electric field waveform with a frequency of 1.05KHz, and the orange line represents the electric field waveform with a frequency of 1.00KHz, the red line represents the electric field waveform with a frequency of 50Hz).

  • According to the electrical circuit representing membrane (Hodgkin-Huxley), the nerve cell membrane has low-pass filtering property, so no action potential will be generated for high-frequency current stimulation. And in the 2017 Cell magazine, the section on the principle of temporal interference stimulation also provided relevant descriptions in this regard. (Dmochowski J, Bikson M. Noninvasive neuromodulation goes deep[J]. Cell, 2017, 169(6): 977-978.)

We have made the following modifications to address the issue of inappropriate discussion above:

Original [56-]:

Temporal interference transcranial magnetic stimulation is a simulation method similar to temporal interference transcranial electrical stimulation [8-9]. The inner stimulation target superimposes a low-frequency envelope induction electric field, and the area where the non-target target is located is a high-frequency signal. Due to the low-pass characteristic of the neuron membrane, the neurons in the area where the envelope is not generated will not respond, thereby activating the deep part of the brain, the role of local area neurons.

Modified version [58-]:

According to references [9-15], both temporal interference transcranial magnetic stimulation and temporal interference transcranial electrical stimulation are deep brain stimulation schemes based on the principle of temporal interference. Adjusting the spatial position of the coil can generate a low-frequency envelope-induced superimposed electric field in the target area, making neurons follow this low-frequency envelope oscillation to generate electrical activity; In nontarget areas, superimposed induction electric fields are dominated by high-frequency induction electric fields, and combined with the extensively accepted low-pass filtering characteristic mechanism of neuronal membranes, the neural electrical activity in this area will not be able to follow very-high-frequency oscillating electric fields[13].

  1. 74-75 unintelligible cFOS description

In response to the unclear description of C-Fos, we have provided additional explanations aimed at readers' understanding of the role of this marker. C-Fos is an indirect marker of neural activity, and it is commonly used for electrical and magnetic stimulation of gene expression in nerve cells.

Original [74-]:

Adam Khalifa et al. designed and optimized a custom solenoid capable of generating temporally interfering induced electric fields to stimulate the rodent brain and express neurons for tracing through the C-Fos activation situation [12]. The results showed no C-Fos expression in the region affected by only one high-frequency magnetic field, indicating that the recruitment of neural activity in the non-target region was ineffective. In contrast, the region subjected to the superimposed low-frequency envelope induced field showed stronger C-Fos expression, indicating that temporal interference TMS could activate neuronal action potentials.

Modified version [88-]:

Adam Khalifa et al. designed and optimized a custom solenoid capable of generating temporally interfering induced electric fields to stimulate the rodent brain and express neurons for tracing through the C-Fos activation situation [14]. The results showed no C-Fos expression in the region affected by only one high-frequency magnetic field, indicating that the recruitment of neural activity in the non-target region was ineffective. In contrast, the region subjected to the superimposed low-frequency envelope-induced field showed stronger C-Fos expression, indicating that temporal interference TMS could activate neuronal action potentials. In 1986, Greenberg et al. first discovered the expression of C-Fos within a few minutes after activating nerve cells. Utilizing this characteristic, C-Fos can serve as a marker of neuronal activity, it typically used for gene expression in electrical and magnetic stimulation of nerve cells at present [15].

  1. 169-177 tasks are explained in an unclear way, also using a bad writing form as occurs often in the text, please avoid using lists of "imperative" sentences.

Thank you for raising the issue of inappropriate word order and grammar in this description. We have made the following modifications to address the issue of excessive use of imperative sentences

Original [169-]:

There are two main tasks for simulating multi-target temporal interference magnetic stimulation: (1) Select coils 1, 2, and 5 to feed in differential frequency alternating current, and perform simulations in human tissue fluid and human brain sphere to realize multi-target array coils temporal interference: choose coils 1, 5, and 6 to pass in differential frequency alternating current, and simulate in the human tissue fluid model to analyze the relationship between the focus area and the arrangement of the coils. (2) The total current of fixed coils 1 and 2 is 4000A, the current frequency of coil 1 is kept at 1.05KHz, and the current frequency of coil 2 is 1.00KHz. By adjusting the current ratio of the two coils, observe the change in the peak position of , and analyze the relationship between the movement of the focus area and the current ratio of coils 1 and 2.

Modified version [215-]:

There are two main tasks for simulating multi-target temporal interference magnetic stimulation: (1) Implementation of multi-target array coil temporal interference: This paper conducted temporal interference simulations of two sets of physiological saline models for multi-target array coils, with the first set being conducting coils 1, 2, and 5; The second group is the conduction coils 1, 5, and 6. In addition, in the ball model, the distribution of the focusing area was analyzed when coils 1, 2, and 5 were turned on. (2) Analyze the relationship between the current ratio of two different frequency coils and the movement of the focusing area: apply a 1KHz AC signal to coil 1 and a 1.05KHz AC signal to coil 2, adjust their current ratio, and observe the changes in the focusing area.

  1. 185 what is the aim of this cuboid? What does it model? Why these dimensions?

The rectangular model is designed to simulate a human tissue fluid model, which can perform qualitative analysis of multi target temporal interference stimuli, and has also been used for modeling and analysis in previous studies.The reason for choosing the saline cuboid model is described in the supplementary article:

Original [185-]:

All simulations are carried out on the simulation platform COMSOL, a multi-physics coupling finite element simulation software that includes multiple modules such as mechanics, electromagnetics, acoustics, and heat transfer. For the simulation of temporal interference transcranial magnetic stimulation, the electromagnetic field module of COMSOL is selected for finite element analysis. The specific steps are (1) Two physical models are built, one of which is the temporal interference multi-target magnetic stimulation saline model, which is used to simulate Human tissue fluid-induced electric field distribution, including a cuboid with a size of 300×300×40mm and an array coils;

Modified version [225-]:

All simulations are carried out on the simulation platform COMSOL, a multi-physics coupling finite element simulation software that includes multiple modules such as mechanics, electromagnetics, acoustics, and heat transfer. For the simulation of temporal interference transcranial magnetic stimulation, the electromagnetic field module of COMSOL is selected for finite element analysis. The specific steps are (1) Two physical models are built, one of which is the temporal interference multi-target magnetic stimulation saline model, which is used to simulate human tissue fluid-induced electric field distribution, including a cuboid with a size of 300×300×40mm and an array coils, the reason for choosing this size for the cuboid is that: the firstly, if the size is too large, it will occupy more computing resources and prolong the calculation time; secondly, the model size should not be smaller than the peripheral size of the array coil to prevent the induced electric field distribution area from not representing the spatial distribution of the electric field of the array coil; finally, considering that deep brain stimulation usually occurs in the white matter and gray matter regions within the range of 15-35mm of the brain, combined with the above reasons, this size positioning selected 300×300×40mm[36-38].

As shown in the following literature:

[36] Chunye, R., et al. (1995). "A novel electric design for electromagnetic stimulation-the Slinky coil." IEEE Transactions on Biomedical Engineering 42(9): 918-925.

[37] Afuwape, O. F., et al. (2021). "Measurement and Modeling of the Effects of Transcranial Magnetic Stimulation on the Brain." IEEE Transactions on Magnetics 57(2): 1-5.

[38] de Lara, L. I. N., et al. (2021). "A 3-axis coil design for multichannel TMS arrays." NeuroImage 224: 117355.

  1. 187 the sphere: what is the depth of each layer? Is it based on a realistic model of the brain? of course not. Why is it not possible to use a realistic brain model which can be obtained by other sources and open source software?

Thank you for your question about "why choose a ball model instead of a real brain model", we will discuss in detail the reasons why the five layer sphere model was chosen to replace the human brain model in the article, and we describe in more detail the source of the electrical characteristic parameters used in the model:

Original [186-]:

the second is a temporal interference multi-target magnetic stimulation sphere model, which has five layers, including gray matter, white matter, cerebrospinal fluid, skull, and the scalp can be used to simulate the induced electric field distribution in the human brain. (2) Assign material properties. The conductivity of the saline model is set to 0.333S/m, and the relative permittivity is 100; the conductivity of the white matter is set to 0.062 S/m, and the relative permittivity is 69811; the conductivity of the gray matter is set to 0.098 S/m, relative permittivity is 69811; cerebrospinal fluid conductivity is set to 2 S/m, relative permittivity is 109; skull conductivity is set to 0.021 S/m, relative permittivity is 2702; scalp conductivity is set to 0.307 S/m, the relative permittivity is 149050; the conductivity of the array coils is set to 5.998×107S/m, and the relative permittivity is 1 [19]. (3) Add physical field-electromagnetic field to solve Maxwell's equations numerically. (4) Use the transient researcher to solve the time-domain distribution of the induced electric field intensity   of the x component, and calculate the low-frequency envelope induced electric field intensity  and modulation intensity  at any position n.

Modified version [240-]:

The other is a temporal interference multi-target magnetic stimulation sphere model, which has five layers, including scalp, skull, cerebrospinal fluid (csf), and gray matter can be used to simulate the induced electric field distribution in the human brain. The scalp, skull, cerebrospinal fluid (csf), and gray matter were 4.9mm, 7.4mm, 2.5mm, and 7.2mm, respectively. The innermost white matter of the brain was a sphere with a radius of 70mm[39-40]. Here, it should be emphasized that the reason for choosing the five layer sphere model instead of the real human brain model in this article is as follows: the concentric sphere model has been used in multiple literature[9,14,31,41], with high accuracy and little difference from the real head model. It can be used for qualitative and quantitative analysis to a certain extent; Secondly, using a real brain model, such as MRI segmentation of images, will generate a large number of units. Although many articles on coil optimization design and analysis of induced electric field distribution use real brain models for simulation, transient simulations with small steps such as time interference (usually within 0.1ms) consume more computer resources.(2) Assign material properties. The conductivity of the saline model is set to 0.333S/m, and the relative permittivity is 100; the conductivity of the white matter is set to 0.062 S/m, and the relative permittivity is 69811; the conductivity of the gray matter is set to 0.098 S/m, relative permittivity is 69811; cerebrospinal fluid conductivity is set to 2 S/m, relative permittivity is 109; skull conductivity is set to 0.021 S/m, relative permittivity is 2702; scalp conductivity is set to 0.307 S/m, the relative permittivity is 149050; the conductivity of the array coils is set to 5.998×107 S/m, and the relative permittivity is 1. The electrical parameters of human tissue used in this paper are provided by the online database of IFAC in the United States [42]. This database provides data on the electrical properties of different tissues under the influence of electromagnetic fields from 10 Hz to 100 GHz, which is more targeted than other conductivity data [43].

  1. 290 orphan sentence

Thank you for raising the issue of statement errors. We have made the following modifications:

Original [290-]:

The analysis of the above two models shows that the designed array coils can realize multi-target magnetic stimulation, and the  of the low-frequency envelope induced electric field at any target position is related to the position and the amplitude of the two high-frequency induced electric fields. value dependent.

Modified version [394-]:

The analysis of the above two models shows that the designed array coils can realize multi-target magnetic stimulation, and the  of the low-frequency envelope induced electric field at any target position is related to the position and the amplitude of the two high-frequency induced electric fields.

  1. 292-298 not intelligible

Thank you for your question regarding the description of the superiority of amplitude modulation depth. Sorry, due to our negligence in writing, the reader is unable to understand this paragraph. Therefore, we have provided detailed explanations in the text.

Original [288-]:

The analysis of the above two models shows that the designed array coils can realize multi-target magnetic stimulation, and the  of the low-frequency envelope induced electric field at any target position is related to the position and the amplitude of the two high-frequency induced electric fields. value dependent. Definition The superiority of  refers to the quality of the envelope degree of the induced electric field time-domain curve. According to the formula (1), it is related to the low-frequency envelope induced electric field modulation intensity, which is determined by the spatial position of the target point, which is reflected in Two aspects:

(a)  The size of determines the upper limit of .

(b) determines the superiority of , that is, under certain circumstances of , the smaller , the better the envelope degree of the induced electric field and the stronger the modulation intensity high, and vice versa.

Modified version [394-]:

The analysis of the above two models shows that the designed array coils can realize multi-target magnetic stimulation, and the  of the low-frequency envelope induced electric field at any target position is related to the position and the amplitude of the two high-frequency induced electric fields. Definition the superiority of  refers to the quality of the envelope degree of the induced electric field time-domain curve. In theory, the higher the amplitude modulation intensity, the overall amplitude of the waveform of the superimposed induced electric field is close to that of the low-frequency induced electric field, which can cause neural action. According to the equation (2), it is related to the low-frequency envelope induced electric field amplitude modulation intensity  , which is determined by the spatial position of the target point, which is reflected in two aspects:

  • According to equation 2, keep the excitation source unchanged, the upper limit of is related to  in any point, and the maximum value of  is .(when  ).
  • determines the superiority of , that is, keep the excitation source unchanged, in any point, the lower the , the better the envelope degree of the induced electric field and the stronger the amplitude modulation intensity high.
  1. 320 should it be 1:1?

Thank you for raising the issue of incorrect current ratio in the article. We have made the following changes

Original [320-]:

Each array has two sets of data, where data 1 represents the experimental group (1:3, the broken line marked by a black triangle in Figure 6B, 6C), and data 2 represents the control group (1:3, the red circle in Figure 6B, 6C Marked line), draw the two groups of data of array 1 and array 2 in two charts for analysis.

Modified version [296-]:

set the current ratio of the Group 1 to 1:1, that is, both coils are connected to 2000A ; set the current ratio of the Group 2 to 1:5, that is, the coil 1 is connected to 3333A, and the coil 2 is connected to 667A for simulation analysis.

  1. 319 why using experimental and control? These are not experiments or experimental groups rather different simulating conditions

We have made the following modifications to the inappropriate vocabulary in the control and experimental groups, specifically changing the control group to Group 1 and the experimental group to Group 2.

Original:

307: to 4000A, set the current ratio of the control group to 1:1, that is, both coils are connected

337: Figures 6B and 6C is smaller than that of the control group, which this study will describe

425: array 1 and array 2 is smaller than that of the control group (1:1) because the simulation

429: the control group. In summary, this paper simulates the relationship between the

308:to 2000A ; set the current ratio of the experimental group to 1:3, that is, the coil 1 is

336:ever, the experimental group's overall induced electric field intensity in the two arrays in

424:found that the overall induced electric field modulus of the experimental group (1:3) in

Modified version:

296: currents of the two coils to 4000A, set the current ratio of the Group 1 to 1:1,

337: intensity in the two arrays in Figures 7B and 7C is smaller than that of the Group 1,

547: smaller than that of the Group 1 (1:1) because the simulation in Section 3.3 is based on

550: resulting in the overall intensity being smaller than that of the Group 1. In summary,

297:coils are connected to 2000A ; set the current ratio of the Group 2 to 1:5, that is, the coil 1

439:shifted by 45mm. However, the Group 2's overall induced electric field

546:overall induced electric field modulus of the Group 2 (1:5) in array 1 and array 2 is

  1. 410 orphan sentence

Thank you for raising the issue of statement errors. We have made the following modifications

Original [410-]:

At the same time, experiment 2 shows that adjacent coils will not add extra focal areas, in the next work, change the size of the coil, and expand the number of coils to N in a clock-wise direction to achieve temporal interference magnetic stimulation for more targets; on the other hand, although coil No. The focal zone fine-tuning technique mentioned in Section 2 optimizes the coil current ratio, and the position of the focal zone can be adjusted precisely.

Modified version [530-]:

At the same time, the second group shows that adjacent coils will not add extra focal areas, in the next work, change the size of the coil, and expand the number of coils to n in a clockwise direction to achieve time-interference magnetic stimulation for more targets. In addition, the focal zone fine-tuning technique mentioned in Section 3.3 optimizes the coil current ratio, and the position of the focal zone can be adjusted precisely.

  1. Figure 4C and 5: details are not appreciable, especially the location and text of point in circles. Colormaps miss the units.

Thank you for your valuable feedback on the chart. Considering that your proposed stimulation depth of 5-10 will limit deep brain stimulation, we have extended the original stimulation depth from 5mm to 20mm, which is 25mm away from the coil.

We have made the following modifications:

Original graph:

Figure 4C                                   Figure 4H 

Figure 5B                                   Figure 7A

Modified version:

Figure 4C                                   Figure 4H 

Figure 6B                                   Figure 9A

  1. What is the depth of these points? A 3D representation of the simulation map should be provided to locate the real depth of fields of significant intensity.

Regarding question 12 above, we have carefully considered your opinion and selected points that are 25mm away from the coil, meaning that all points are within the plane of a 20mm deep physiological saline model or sphere model.

In response to your 3D map of AM stimulation depth, we have provided the following diagram to represent the longitudinal plane view located in coil 1, coil 2, and coil 5, where this plane is segmented along the axis of the three coils , (the y-axis in Figure 4A)

We supplemented this section in the article after line 361:

Supplementary content:

For the depth distribution of induced electric field, the z-direction segmentation is carried out along the y-axis in Figure 4A to obtain the depth induced electric field distribution of coils 1, 2 and 5 , in physiological saline. The results are shown in Figure 5A and Figure 5B. Figure 5A shows the induced electric field distribution no temporal interference, while Figure 5B shows the  distribution under temporal interference. By comparison, it was found that as the depth increased, the focusing area under temporal interference was significantly better than that of the group without time interference. Although the focus area is mainly concentrated at a depth of 10mm, it may not reach the gray area. The reason is that this simulation uses a qualitative simulation analysis at a frequency of 1KHz. To stimulate deeper areas, researchers can increase the frequency of the differential signal to the level of 10KHz, as used in reference [14].

Figure 5 Depth distribution map of induced electric field (A) Distribution of deep induction electric field no temporal interference. (B) Distribution of  temporal interference

  1. Figure 4F wrong points in legends (1), figure 4E in legend states two times the same point.

Thank you again for your question. We have made modifications to address the numbering issues in these two images. All selected points here are slightly different from the original text, and the selection method is the same as question 12. All points are moved down 15mm from the original text

Original graph:

Figure 4F                                   Figure 4E 

Modified graph:

Figure 4F                                   Figure 4E

  1. In Figure 4A and G coil number should be indicated

Thank you for your question about number numbering. We have made the following modifications:

Original graph:

Figure 4A                               Figure 4G

Figure 5A   

Modified graph:

Figure 4A                                   Figure 4G

Figure 6A 

  1. Figure 4C and 5B: line 274 states that only two focal spots are produced, but 4 are visible. Two additional focus points are presente (red in the map) at the extreme sides and should be analyzed and discussed.

Thank you for raising the four focal spots issues. We have conducted a detailed analysis during the discussion, as follows:

Original [214-]

Select the upper surface of the normal saline to be 5mm, that is, the distance from the lower surface of the coil is 10mm, draw the induced electric field of the x component as a plane map, and select the moment when the modulus is maximum. The position of the first group of simulation experiment coils is shown in Fig. 4A and Fig. 4B. It is stipulated that the center of the central coil is the coordinate origin, the line connecting the centers of coils 1 and 2 is the y-axis, the positive direction points to coil 2, the x-axis is a straight line perpendicular to the y axis, and the positive direction points to coils 3 and 4. Figure 4C has two focal areas located between coils 1 and 2 and coils 1 and 5, respectively, which meet the needs of multi-target stimulation. In addition, there are two areas with lower peak values. The reasons for this will be discussed in Section 4 of this article.

Modified version [301-]

Select the upper surface of the normal saline to be 20mm, that is, the distance from the lower surface of the coil is 25mm, draw the induced electric field of the x component as a plane map, and select the moment when the modulus is maximum. The position of the first group of simulation experiment coils is shown in Fig. 4A and Fig. 4B. It is stipulated that the center of the central coil is the coordinate origin, the line connecting the centers of coils 1 and 2 is the y-axis, the positive direction points to coil 2, the x-axis is a straight line perpendicular to the y axis, and the positive direction points to coils 3 and 4. Figure 4C has two focal areas located between coils 1 and 2 and coils 1 and 5, respectively, which meet the needs of multi-target stimulation. In addition, there are two weaker focusing regions located on the outer edges of coil 5 and coil 2, respectively. The reason for its occurrence is related to the current flowing through the outer edge of the two coils. At the same time, this article selects the direction of the induced electric field as the x direction, so the focus area is concentrated near the y-axis. Despite the presence of these two weakly focused regions, coils 2 and 5 are connected with continuous high-frequency stimulation signals, so there is no cortical response in this region.

Original [270-]

The results showed that there were two focal regions in Figure 5A、5B, and the positions were similar to the focal region distribution of the saline model.

Modified version [383-]

The results showed that there were two focal regions in Figure 6A、6B, and the positions were similar to the focal region distribution of the saline model. Similar to the physiological saline rectangular model, the spherical model still has two weak focusing regions located on the outer edges of coil 5 and coil 2, respectively. The reason for its occurrence is related to the current flowing through the outer edges of the two coils. Similarly, coils 2 and 5 are connected with continuous high-frequency stimulation signals, so there is no cortical response in this area.

  1. Figure 5D a(1:3) plot is not coherent with AM plot in Figure 5B

Thank you for pointing out the error in Figure 5. Due to the shorter focus area movement in the original text with a current ratio of 1:3, we have adjusted the current ratio to 1:5 , and restated this section:

Original [302-]:

In the normal saline simulation model, select coil 1 and coil 2 to analyze the relationship between the deviation of the peak position of  and the current ratio of the two coils, and keep the current frequency of coil 1 at 1.05KHz and the current frequency of coil 2 at 1.00KHz. The specific method is: to keep the current frequency of coil 1 at 1.05KHz and the current frequency of coil 2 at 1.00KHz, fix the sum of the currents of the two coils to 4000A, set the current ratio of the control group to 1:1, that is, both coils are connected to 2000A ; set the current ratio of the experimental group to 1:3, that is, the coil 1 is connected to 1000A, and the coil 2 is connected to 3000A for simulation analysis.

Same as Section 3.1, draw a plane parallel to the coil at 5mm below the upper surface of the normal saline model, that is, a distance of 10mm from the lower surface of the coil, and draw the x component of the induced electric field. In Fig. 6A, the line connecting the centers of coils 1 and 2 is first defined as the y-axis, and the positive direction points to coil 2. Secondly, the midpoint of the line connecting the centers of the two coils is selected as the origin of the coordinates. Finally, the x-axis is defined as a straight line passing through the origin and perpendicular to the y-axis. The direction points to coils 3, and 4 directions. On the x-axis (x=0), select 10 points on the left and right sides with the origin as the center, and the step length is 2mm, which is called array 1 (Figure 6B); similarly, move array 1 positively along the x-axis by 5mm, which is Array 2 (Fig. 6C). `

Figure 6. The simulation results of the movement of the focus area (A) Schematic diagram of coil placement (B) Array 1 point line graph vs. line graph, the broken line marked with black triangles indicates 1:3, and the red circle marked line indicates 1:1 (C ) Array 2-point line diagram vs. line diagram, the broken line marked with black triangles indicates 1:3, and the line marked with red circles indicates 1:1 (D) The induced electric field distribution at point a at the ratio of 1:1 and 1:3 (E) Induced electric field distribution at two points c and d at a ratio of 1:3 (F) Distribution of an induced electric field at two points c and e at a ratio of 1:3.

Figure 6B shows that the dot plot lines of data 1 and 2 are symmetrical, and the peak of the induced electric field  is located at the center of the symmetry and gradually decays along the left and right, which is similar to the distribution of the induced electric field in the figure-eight focus area. In addition, the symmetry centers of  of the two data show that there is no coincidence, and the distance is 8mm (the peak is at the origin of the coordinates in the case of 1:1, and the peak is at the coordinate (-8) in the case of 1:3). The results show that changing the current ratio will affect the peak position of  , and the peak position shifts to the side with smaller current, which supports the formula (2). Excluding chance, the simulation of array 2 was carried out, and the result was similar to that of array 1. As shown in Fig 6C, the peak position of  shifted by 4mm. However, the experimental group's overall induced electric field intensity in the two arrays in Figures 6B and 6C is smaller than that of the control group, which this study will describe in the discussion in Section 4.

To further observe the overall situation of the low-frequency induced electric field envelope, select 5 points on the y-axis, namely a (0, 0), b (0, -4), c (0, -6), d (0, -2) and e(0,6). Fig 6D shows the envelope of point an at the ratio of 1:1 and 1:3, where the blue line represents 1:1. The green line represents 1:3. The results show that the envelope of the blue line is obviously better than that of the green line. There is a certain deviation of the origin when the ratio is 1:3. Both Figure 6E and Figure 6F are compared at a ratio of 1:3. In Figure 6E, the blue line represents c, and the green line represents d; in Figure 6F, the blue line represents c, and the green line represents e. Through comparison, it is found that point c and point d envelope-induced electric field is similar, but the difference between point c and point e is relatively large. The reason is that when the current ratio changes from 1:1 to 1:3, the center of symmetry (that is, the peak value of ) shifts, resulting in the focus area moving towards the negative direction of the y-axis, so the envelope electric field waveform at point e is significantly worse than that at point c.

In summary, the peak shift of  conforms to formula (2), that is, the overall focus area moves toward the side with a weak current. The reason is: Assume that the distribution of the two coils on the y-axis is consistent, that is, keep the sum of at any point on the y-axis unchanged (this is based on the symmetry of the centers of coils 1 and 2 concerning the x-axis), so the peak of  appears at the maximum value of formula (2), that is, when . Based on the above theory, when coil 2 is fed with a current of 3000A, which is greater than the 1000A current of coil 1, the induced electric field strength  near coil 2 is greater than the induced electric field strength  near coil 1, so the point of  will be biased and move to the vicinity of coil 1, the side where the current is weak.

Original [412-]:

The simulation model was built with physiological saline to analyze the relationship between the change of the focus area of temporal interference magnetic stimulation and the current ratio. Figure 6B shows that when the ratio is changed from 1:1 to 1:3, the focus area is shifted from the center of the coil, that is, the origin to 8mm, and there is still symmetry, indicating that the current adjustment ratio can be shifted from the peak position of  , thereby changing the position of the overall focus area. In addition, it can be observed that on the y-axis of x=5, array 2 is selected to perform simulation analysis again, and it is found that the peak offsets of the two data  are both smaller than x=0. The author believes that the reason is that the coordinate points of the two data in array 2 are far away from the centers of the two coils, so the amplitude of the induced electric field  is relatively small overall. But in general, data 2 of the two arrays has  peak shift, and the overall focus area has moved. From Figures 6B and C, this study also found that the overall induced electric field modulus of the experimental group (1:3) in array 1 and array 2 is smaller than that of the control group (1:1) because the simulation experiment in Section 3.3 is based on the sum of the two coils maintaining. Therefore, when the ratio is changed, the focus area tends to move towards the direction of the low induced electric field intensity, resulting in the overall intensity being smaller than that of the control group. In summary, this paper simulates the relationship between the movement of the focus area of temporal interference magnetic stimulation and the current ratio and selects two coils with different frequencies for verification and analysis. The author believes that the key to fine-tuning the focus lies in the ratio of the induced electric field of different frequencies in the focus area, rather than the number of activated coils, so this paper does not conduct experiments with multiple coils (the combination of multiple coils at two frequencies is essentially a superposition of fields).

Modified version [291-]:

In the normal saline simulation model, select coil 1 and coil 2 to analyze the relationship between the deviation of the peak position of  and the current ratio of the two coils, and keep the current frequency of coil 1 at 1.00KHz and the current frequency of coil 2 at 1.05KHz. The specific method is: to keep the current frequency of coil 1 at 1.00KHz and the current frequency of coil 2 at 1.05KHz, fix the sum of the currents of the two coils to 4000A, set the current ratio of the Group 1 to 1:1, that is, both coils are connected to 2000A ; set the current ratio of the Group 2 to 1:5, that is, the coil 1 is connected to 3333A, and the coil 2 is connected to 667A for simulation analysis.

Modified version [419-]:

Same as Section 3.1, draw a plane parallel to the coil at 20mm below the upper surface of the normal saline model, that is, a distance of 25mm from the lower surface of the coil, and draw the x component of the induced electric field. In Fig. 7A, the line connecting the centers of coils 1 and 2 is first defined as the y-axis, and the positive direction points to coil 2. Secondly, the midpoint of the line connecting the centers of the two coils is selected as the origin of the coordinates. Finally, the x-axis is defined as a straight line passing through the origin and perpendicular to the y-axis. The direction points to coils 3, and 4 directions. On the x-axis (x=0), select 15 points on the left and right sides with the origin as the center, and the step length is 5mm, which is called Array 1 (Figure 7B); similarly, move Array 1 positively along the x-axis by 5mm, which is Array 2 (Figure. 7C).

Figure 7. The simulation results of the movement of the focus area (A) Schematic diagram of coil placement (B) Array 1: the dotted line diagram at x=0, represented by a black line marked with a box for 1:1, and a red line marked with a circle for 1:5 (C ) Array 2: the dotted line diagram at x=5, represented by a black line marked with a box for 1:1, and a red line marked with a circle for 1:5 (D) The induced electric field distribution at point a at the ratio of 1:1 and 1:5,when x=0 (E) Induced electric field distribution at points b at a ratio of 1:5, when x=0.

Figure 7B shows that the dot plot lines of data 1 and 2 are symmetrical, and the peak of the induced electric field  is located at the center of the symmetry and gradually decays along the left and right, which is similar to the distribution of the induced electric field in the figure-eight focus area. In addition, the symmetry centers of  of the two data show that there is no coincidence, and the distance is 45mm (the peak is at the origin of the coordinates in the case of 1:1, and the peak is at the coordinate (45mm) in the case of 1:5). The results show that changing the current ratio will affect the peak position of , and the peak position shifts to the side with smaller current, which supports the formula (2). Excluding chance, the simulation of array 2 was carried out, and the result was similar to that of array 1. As shown in Fig 7C, the peak position of  shifted by 45mm. However, the Group 2's overall induced electric field intensity in the two arrays in Figures 7B and 7C is smaller than that of the Group 1, which this study will describe in the discussion in Section 4.

To further observe the overall situation of the low-frequency induced electric field envelope, select 2 points on the y-axis, namely a (0, 0), b (0, 45). Fig 7D shows the envelope of point a at the ratio of 1:1 and 1:5, where the black line represents 1:1. The green line represents 1:5. The results show that the envelope of the black line is obviously better than that of the green line. There is a certain deviation of the origin when the ratio is 1:5. Fig 7E shows the envelope of point b at the ratio of 1:1 and 1:5, where the gray line represents 1:1. The red line represents 1:5. The results show that the envelope of the red line is obviously better than that of the gray line. This is because in the 1:5 situation, the focus area shifted.

Modified version [545-] :

The simulation model was built with physiological saline to analyze the relationship between the change of the focus area of temporal interference magnetic stimulation and the current ratio. Figure 7B shows that when the ratio is changed from 1:1 to 1:5, the focus area is shifted from the center of the coil, that is, the origin to 45mm, and there is still symmetry, indicating that the current adjustment ratio can be shifted from the peak position of , thereby changing the position of the overall focus area.The offset of the region is consistent with the magnitude mentioned in the literature [12], both at the centimeter level. From Figures 7B and C, this study also found that the overall induced electric field modulus of the Group 2 (1:5) in array 1 and array 2 is smaller than that of the Group 1 (1:1) because the simulation in Section 3.3 is based on the sum of the two coils maintaining. Therefore, when the ratio is changed, the focus area tends to move towards the direction of the low induced electric field intensity, resulting in the overall intensity being smaller than that of the Group 1. In summary, this paper simulates the relationship between the movement of the focus area of temporal interference magnetic stimulation and the current ratio and selects two coils with different frequencies for verification and analysis. The author believes that the key to fine-tuning the focus lies in the ratio of the induced electric field of different frequencies in the focus area, rather than the number of activated coils, so this paper does not conduct simulation with multiple coils (the combination of multiple coils at two frequencies is essentially a superposition of fields).

  1. 442-447, please rephrase: these are too ambitious aims and present form is used which seems to refer to work in progress.

Thank you for your ambitious description in the article. We deeply apologize for this. Currently, we have made changes to the inappropriate statements you mentioned. Thank you again for your review

Original (442-) :

For future work, the author believes that the precise adjustment of the focus area position can be achieved by optimizing the coil structure through intelligent algorithms, de-signing the coil space topology, and adjusting the ratio of different frequency currents. In addition, by collecting multi-channel EEG signals and MRI images of patients before and after stimulation, and based on the characteristics of behavioral scale-EEG-MRI image multimodal data fusion, a TMS electromagnetic closed-loop control scheme is designed. Based on the clinical effect, the precise control technology is investigated to improve the effectiveness and safety of the patient's motor function, and the clinical effect of the intelligent magnetic stimulation system is evaluated. Based on the clinical effect, the precise control technology is investigated to improve the effectiveness and safety of the patient's motor function, and the clinical effect of the intelligent magnetic stimulation system is evaluated.

In this study, we proposed a multi-target temporal interference magnetic stimulation system with array coils, a multi-brain area co-stimulation technology that integrates temporal interference, which can simultaneously stimulate multiple targets in deep brain regions. In this study, we proposed a multi-target temporal interference magnetic stimulation system with array coils, a multi-brain area co-stimulation technology that integrates temporal interference, which can simultaneously stimulate multiple targets in deep brain regions. We validated this technique in saline and human brain sphere models. In addition, we adopted the method of shifting the focus area without moving the coil by changing the current ratio between coils with different frequencies and verified the feasibility of this method through simulation. These results indicate that the multi-target temporal interference magnetic stimulation with the array coils can non-invasively stimulate multiple network nodes in the brain simultaneously, and precise positioning can be achieved by controlling the conduction of different coils and changing the proportion of current in the coils.

Modified version (564-):

For future work, the author believes that the precise adjustment of the focus area position can be achieved by optimizing the coil structure through algorithms, designing the coil space topology, and adjusting the ratio of different frequency currents. In addition, by collecting multi-channel EEG signals and MRI images of patients before and after stimulation, and based on the characteristics of behavioral scale-EEG-MRI image multimodal data fusion, a TMS electromagnetic closed-loop control scheme is designed. Based on clinical effects, optimizing control technology is expected to improve.

In this study, we propose a multi target temporal interference magnetic stimulation system with array coils, which integrates multi brain region co-stimulation technology with temporal interference, aiming to stimulate multiple targets in the deep brain. We validated this technique in saline and human brain sphere models. In addition, we adopted the method of shifting the focus area without moving the coil by changing the current ratio between coils with different frequencies and verified the feasibility of this method through simulation. These results indicate that the multi-target temporal interference magnetic stimulation with the array coils can non-invasively stimulate multiple network nodes in the brain simultaneously, and precise positioning can be achieved by controlling the conduction of different coils and changing the proportion of current in the coils.

  1. Title: nerve is misleading (the aim is brain stimulation here). Intelligent optimization is misleading since there is no optimization in this manuscript and no use of AI. Also remove any reference in the text to these aspects (intelligent is often used).

Thank you for your inappropriate statement about the word "intelligent" in the title "Intelligent Optimal Design of Array Coils for Multi-target Electromagnetic Nerve Stimulation System", we have changed the title based on your review, and reviewed the full text to modify the sentence containing "intelligent"

original title:

Intelligent Optimal Design of Array Coils for Multi-target Electromagnetic Nerve Stimulation System

Original (100-) :

This paper introduces the concept of multi-brain area co-stimulation, a new brain stimulation technique designed to simultaneously stimulate multiple nodes of a brain network in deep brain regions. Specifically, a multi-coil array structure was designed, which can drive the conduction of different coils to simultaneously stimulate multiple targets through an intelligent algorithm[18-22].Moreover, rough positioning can be performed by controlling the conduction of different coils; fine-tuning the position by adjusting the current ratio of the conduction coils to achieve precise stimulation of multiple brain regions and targets. It is expected to be used for synergistic stimulation of multiple brain regions to treat different diseases and regulate brain network functions precisely.

Original (442-) :

For future work, the author believes that the precise adjustment of the focus area position can be achieved by optimizing the coil structure through intelligent algorithm, designing the coil space topology, and adjusting the ratio of different frequency currents. In addition, by collecting multi-channel EEG signals and MRI images of patients before and after stimulation, and based on the characteristics of behavioral scale-EEG-MRI image multimodal data fusion, a TMS electromagnetic closed-loop control scheme is designed. Based on clinical effects, optimizing control technology is expected to improve

Modified title:

Optimal Design of Array Coils for Multi-target Adjustable Electromagnetic Brain Stimulation System

Modified version (130-):

This paper introduces the concept of multi-brain area co-stimulation, a new brain stimulation technique designed to simultaneously stimulate multiple nodes of a brain network in deep brain regions. Specifically, a multi-coil array structure was designed, which can drive the conduction of different coils to simultaneously stimulate multiple targets through an algorithm[18-22].Moreover, rough positioning can be performed by controlling the conduction of different coils; fine-tuning the position by adjusting the current ratio of the conduction coils to achieve precise stimulation of multiple brain regions and targets. It is expected to be used for synergistic stimulation of multiple brain regions to treat different diseases and regulate brain network functions precisely.

Modified version (564-):

For future work, the author believes that the precise adjustment of the focus area position can be achieved by optimizing the coil structure through algorithms, designing the coil space topology, and adjusting the ratio of different frequency currents. In addition, by collecting multi-channel EEG signals and MRI images of patients before and after stimulation, and based on the characteristics of behavioral scale-EEG-MRI image multimodal data fusion, a TMS electromagnetic closed-loop control scheme is designed. Based on clinical effects, optimizing control technology is expected to improve.

  1. Limitations must be discussed including the positioning of the device on the skull, the possibility to scale the stimulation locations to any skull or brain size etc.

Thank you for your question about the size and position of the coil above the skull, etc. We have supplemented it in the text according to your request.

Original [128-]:

All the coils are circular coils of the same size, and the arrangement is shown in Figures 3A and 3B. The center of the circular coil numbered 1 coincides with the center of the array, which is called the center coil; then the remaining numbered coils (from small to large) are arranged clockwise around the center coil. The coil radius is 25mm, the wire diameter is 2mm, the number of turns is 6, and the material is metallic copper with good conductivity.

Modified version [162-]:

All coils were circular coils of the same size, arranged as shown in Figures 3A and 3B. The center of the circular coil numbered 1 coincides with the center of the array, which is called the central coil; the remaining numbered coils (from small to large) are arranged clockwise around the central coil. The selection of radius has two factors: firstly, considering that the diameter of the five-layer sphere model is 184mm, the head circumference of adults is between 540-580mm, and there are at least three circles in the same column; Secondly, the coil radius is approximately positively correlated with the stimulus intensity. Specifically, as the coil radius increases, the cortical stimulus intensity gradually increases. So the coil unit is selected with an outer radius of 25mm [29]. The wire diameter is 2mm, the number of turns is 6, and the material is copper with good conductivity. The shell is made of an insulating epoxy resin board, which is characterized by good mechanical properties under medium temperature conditions, stable electrical properties under high humidity conditions, good heat resistance and moisture resistance, and can protect personnel safety.

  1. More importantly, a major limitation of this study refers to:

1) lack of high depth stimulation: the simulations and analyses are performed at 5-10 mm from surface, which is very limited and does not makes any sense in the context of deep-TMS applications, such as sub-cortical nuclei. In addition, no 3D visualization is provided to really understand the position of the focal spots in depth.

2) simultaneous AM and multi-site stimulation is not shown, rather these are only separately discussed, contrary to the main scope of the study. In addition, only specific cases (a few) are discussed, and very few images of simulations are provided, so that that it is difficult to generalize and to appreciate the result. For instance, no 3D colormap representation of AM effect is provided while this would be the most crucial aspect of this study.

Thank you very much for reviewing our article. Regarding your two key suggestions, we will provide the following responses and hope to receive your recognition:

1). As you mentioned, the simulated stimulation depth in our article is only 5-10mm. Based on human anatomical knowledge, this thickness only reaches the skull level and does not stimulate us to the depth of stimulation, such as the gray matter area located below 15mm depth. Based on this, we have redesigned the simulation process to extend the depth to 20mm below the scalp, which is 25mm below the coil. This depth is sufficient to stimulate the gray matter area, and we have redrawn the relevant result graph and conducted a new analysis. In addition, from the result graph, we found that as the depth increases, the intensity of the induced electric field will undergo a certain attenuation. However, we believe that the attenuation can be compensated by increasing the stimulation frequency to 10KHz (based on the fact that an increase in stimulation frequency deepens the stimulation depth). Reference (Khalifa, A., et al. (2023). "Magnetic temporal interference for noninvasive and focal brain stimulation." Journal of neural engineering 20(1): 016002) also provides some support. In addition, the improvement in focusing ability of temporal interference magnetic stimulation compared to traditional magnetic stimulation has been mentioned multiple times in numerous literature, And relevant validation has been conducted, indicating that this technology has great potential in deep brain stimulation.

Figure 5A. Distribution of deep induction electric field no temporal interference

Figure 5B. Distribution of  temporal interference

We supplemented this section in the article after line 361:

Supplementary content:

For the depth distribution of induced electric field, the z-direction segmentation is carried out along the y-axis in Figure 4A to obtain the depth induced electric field distribution of coils 1, 2 and 5 , in physiological saline. The results are shown in Figure 5A and Figure 5B. Figure 5A shows the induced electric field distribution no temporal interference, while Figure 5B shows the  distribution under temporal interference. By comparison, it was found that as the depth increased, the focusing area under temporal interference was significantly better than that of the group without time interference. Although the focus area is mainly concentrated at a depth of 10mm, it may not reach the gray area. The reason is that this simulation uses a qualitative simulation analysis at a frequency of 1KHz. To stimulate deeper areas, researchers can increase the frequency of the differential signal to the level of 10KHz, as used in reference [14].

Figure 5 Depth distribution map of induced electric field (A) Distribution of deep induction electric field no temporal interference. (B) Distribution of  temporal interference

  • Thank you for your questions about multi-target focusing and providing fewer simulated images. We fully accept your comments and have made the following expansions.

We supplemented this section in the article after line 484:

Supplementary content:

Regarding the multi-target simulation problem, two models build in this article (respectively cuboid saline and five-layer ball model) for multi-target correlation analysis. First, in normal saline, two sets of simulations conducted . The purpose is to achieve multi-target focus areas. The selection principle of these two groups of simulation coils is to fully demonstrate the purpose of simulation in this section and to establish a reasonable simulation process. In this regard, in the first group, namely, conduction coils 1, 2, and 5, aimed to investigate whether a multi-target focal area could be formed under temporal interference, although two focal areas , but this result can show that this group of similarly arranged coils in the same column can produce multiple Focus, and there will be no obvious interference; in the second group,  the purpose is to check whether there will be unnecessary interference between adjacent coils, so coils 1, 5 and 6 are selected for analysis, and the results show that although The distance between coils 5 and 6 is relatively close, there is no obvious focus area. The reasons are as follows. The first reason is that the current directions of coils 5 and 6 are the same, resulting in a large attenuation of the induced electric field superposition in the focus area; the second reason is that coils 5 and 6 The same frequency current is passed through so that it will not produce envelope superposition in the intersecting area. To sum up, the first group to explain that the coils in the same column can generate multiple target focus areas, and use the second group to explain whether there is interference between adjacent coils with the same frequency. In addition, after careful consideration in this paper, it is believed that the cross structure in Figure 8A (that is, the cross structure centered on coil 1) and the adjacent coil (same frequency) in Figure 8B is far away from each other and have the same frequency, so the focus area occurs Around the coil 1, that is, the author of this paper believes that the symmetrical arrangement and expansion of the coils can be carried out on the first and second sets of simulations to realize the role of multiple targets.

Figure 8. Schematic diagram of array coil unit arrangemen. (A) The cross structure. (B) Adjacent coil structures with long distances

Reviewer 3 Report

In the introduction, the authors review  recent studies that indicate that brain stimulation using a temporal interference technique might achieve stimulation of deep targets without activating superficial non-target areas.  The technique entails use of two spatially separated coils to generate high-frequency magnetic fields differing in frequency by a small amount. In the brain regions where there is appreciable spatial overlap of the two high frequency fields, interference produces amplitude modulation at a low frequency optimal for neural stimulation.  An investigation by Khalifa et al (2023) using c-fos activation as a marker for neural activation in rodent brains indicates that this technique might be effective.   The authors of this manuscript suggest that at least in principle, simultaneous stimulation of multiple brain regions might be an effect way to modulate neural activity in distributed brain circuits. 

The authors report an investigation of the feasibility of the coordinated stimulation of multiple brain regions.  They used the simulation platform COMSOL to model the effects of temporal interference stimulation achieved by employing multiple surface coils.   They also investigate the possibility that adjusting the relative current strength in a pair of coils might produce an appreciable shift in the  location of the hypothetical neural stimulation, thereby allowing fine tuning of the position of the simulation target.

Unfortunately, it is difficult to understand much of the text on account of poor organization of the material (e.g. the first paragraph of the results section provide a description of the  procedure that would be better located in the methods section) and  multiple grammatical errors (e.g. sentences with the main verb  missing). There also many other types of error. There are errors in the numbering of target sites (e.g. in the text, the discussion of Figure 7 refers to field strength at point 2, while point 2 is not shown in Figure  7; it appears that the text erroneously refers to point 8 as point 2).  In line 320 of the text, the control group are erroneously labelled as (1:3) whereas elsewhere in the text and in Figure 6 it is clear that the control group should be labelled (1:1). In some instances, it is difficult to read the numbers labelling target site (e.g in Figure 4C).  While it is relatively straightforward for the reader to correct many of these errors, even are careful scrutiny, it is difficult to be confident about the authors’ intended meaning.

The validity of the authors’ assumption that the activation occurs when the  direction of the induced electric field is parallel to ‘the axis of the nerve cells’ (meaning undefined) is an oversimplification. The authors justify this assumption with the statement  ‘according to the literature (18)’.  Referecne 18 refers to Roth and Basser’s model of the effect of stimulation a nerve fibre.  Empirical evidence from a concurrent TMS and PET study (Krieg et al 2015 J. Neural Eng. 12 046014 DOI 10.1088/1741-2560/12/4/046014 ) reveals that brain regions activated by the TMS have significantly higher E-field components parallel to pyramidal neurons in the dendrite-to-axon orientation) and in the tangential direction (i.e., parallel to interneurons) at high gradient.

The authors report simulations using two models: a saline cuboid and a five layered sphere model, with layers representing gray matter, white matter, cerebrospinal fluid, skull, and the scalp.  The sphere model is not described in adequate detail (e.g. the thickness of the layers is not specified)  

My greatest misgiving is the limited clinical relevance of the effects reported.  Despite the fact that a major reason for employing temporal interference magnetic stimulation is to achieve stimulation deep within the brain, the authors report the effects of stimulation only at shallow depths. In the saline cuboid model, the effects of stimulation are reported in a plane parallel to the surface and 5 mm below the surface.  For the sphere model, the target plane is described as  10 mm below the coil which is located 5 mm above the highest point of the surface sphere. This implies that target plane is 5mm or less below the surface of the sphere. The authors do not say what tissue type is located at this depth.  One would hope it would be gray matter but if so, the combined thickness of scalp, bone and CSF space is not realistic.

In the case of the manipulation of the relative current in a pair of coils, changing the current ratio from 1:1 to 1:3 produces a  horizontal shift of the peak amplitude modulation that is small compared with the half width of the peak, implying a shift in peak predicted neural excitation is  that is less than the spatial resolution of the stimulation profile. As far as I can see, the magnitude of the reported effects of interest are of minimal clinical relevance.  

The implication that their findings indicate that multi- target temporal interference magnetic stimulation with array coils can simultaneously stimulate multiple network nodes in the brain region in a clinically meaningful manner is not justified.

Author Response

Dear  Reviewer:

Thank you for reviewing our manuscript entitled “Optimal Design of Array Coils for Multi-target Adjustable Electromagnetic Brain Stimulation System” (ID: bioengineering-2314035). Those comments are all valuable and very helpful for revising and improving our paper, as well as the important guiding significance to our researches. We have studied comments carefully and have made correction which we hope meet with approval. The details of the modification are in the attachment.

                                  Author: Dr. Tingyu Wang、Mr. Lele Yan、Dr. Xinsheng Yang、Prof. Duyan Geng、Prof. Dr. Guizhi Xu *、Prof. Alan Wang *

  1. Unfortunately, it is difficult to understand much of the text on account of poor organization of the material (e.g. the first paragraph of the results section provide a description of the  procedure that would be better located in the methods section) and  multiple grammatical errors (e.g. sentences with the main verb  missing). There also many other types of error. There are errors in the numbering of target sites (e.g. in the text, the discussion of Figure 7 refers to field strength at point 2, while point 2 is not shown in Figure  7; it appears that the text erroneously refers to point 8 as point 2).  In line 320 of the text, the control group are erroneously labelled as (1:3) whereas elsewhere in the text and in Figure 6 it is clear that the control group should be labelled (1:1). In some instances, it is difficult to read the numbers labelling target site (e.g in Figure 4C).  While it is relatively straightforward for the reader to correct many of these errors, even are careful scrutiny, it is difficult to be confident about the authors’ intended meaning.

Thank you for raising the issue described in the first paragraph of the results section in the original text. We have made modifications to this issue and adjusted the structure of the article, aiming to address other areas where similar issues arise throughout the entire

added subsection 2.3、2.4 and 2.5:

Modified version [270-]:

2.3 Simulation of multi-target temporal disturbance magnetic stimulation in a saline model

Select physiological saline to simulate human tissue fluid for simulation analysis, and carry out two groups of simulation analysis. The first group of coils 1, 2, and 5 are fed with an alternating current with an amplitude of 2000A, and the current frequency of coil 1 is set to 1.00KHz, and the current of the other coils frequency is 1.05KHz; the second group of coils 1, 5 and 6 is fed with an alternating current with an amplitude of 2000A, the current frequency of coil 1 is set to 1.00KHz, and the current frequency of the other coils is 1.05KHz. The purpose of entering the difference frequency is to be able to modulate a low-frequency 50Hz envelope induction electric field in the focus area. The all array coils are located 5mm from the upper surface of the cuboid. Finally, through COMSOL transient research simulation calculations, the analysis of the focus area in the time domain induction electric field situation.2.4 Simulation of Multi-Target Temporal Disturbance Magnetic Stimulation in a Saline Model

2.4 Simulation of multi-target temporal disturbance magnetic stimulation in spherical model

Select a five-layer sphere model to establish a simulation model of human brain multi-target temporal interference magnetic stimulation. The array coils is located 5mm above the highest point on the sphere's surface, and the rest of the settings are the same as those of normal saline. Select 25mm of the lower surface of the coil as a plane parallel to the coil, which is the 20mm depth of the sphere, and analyze the distribution of the induced electric field of the x component.

2.5 Simulation of the relationship between temporal disturbance magnetic stimulation focal area and current ratio

In the normal saline simulation model, select coil 1 and coil 2 to analyze the relationship between the deviation of the peak position of  and the current ratio of the two coils, and keep the current frequency of coil 1 at 1.00KHz and the current frequency of coil 2 at 1.05KHz. The specific method is: to keep the current frequency of coil 1 at 1.00KHz and the current frequency of coil 2 at 1.05KHz, fix the sum of the currents of the two coils to 4000A, set the current ratio of the Group 1 to 1:1, that is, both coils are connected to 2000A ; set the current ratio of the Group 2 to 1:5, that is, the coil 1 is connected to 3333A, and the coil 2 is connected to 667A for simulation analysis.

Thank you for your inquiry regarding the description of Figures 7 and 4C, as well as some other related issues. We have conducted a full text screening and made the following modifications:

Original [385-]:

This paper analyzes the modulation of the two low-peak focus areas, selects point ⑧ (0, -90) located at the edge of coil 5 in Figure7A as the analysis point, and compares point ① in 3.1, and calculates the x-component of the two points through simulation the time-domain change curve of the induced electric field, the blue line in Figure 7B represents point ①, and the red line represents point ⑧. The results show that the envelope shape of the curve at point ⑧ is worse than point ①, and although the peak value of the induced electric field at the two points differs by 3.27V/m (①: 14.75V/m, ②: 11.48V/m), the  differs by 8.76V/m (①: 11.34V/m, ②: 2.58V/m). The data show that although there are two low-peak focal regions, their amplitude modulation intensity is greatly reduced compared with the high-peak focal region, which is quite different from traditional magnetic stimulation. In the second experiment on the relationship between the focus area and the coil arrangement, the coil arrangement is changed to analyze the influence of adjacent coils on the focus area.

Modified version [522-]

This paper analyzes the modulation of the two low-peak focus areas, selects point ④’ (0, -90) located at the edge of coil 5 in Figure 9A as the analysis point, and compares point ① in 3.1, and calculates the x-component of the two points through simulation the time-domain change curve of the induced electric field, the green line in Figure 9B represents point ①, and the blue line represents point ④’. The results show that the envelope shape of the curve at point ④’ is worse than point ①, and the peak value of the induced electric field at the two points differs by 2.8V/m (①: 6.3V/m, ④’: 3.5V/m), the  differs by 3V/m (①: 4.1V/m, ④’: 1.1V/m). The data show that although there are two low-peak focal regions, their amplitude modulation intensity is greatly reduced compared with the high-peak focal region

Figure 7. (A) ①、④’ focus area  (B) ①、④’ time-domain comparison of induced electric field.

Original [302-]:

In the normal saline simulation model, select coil 1 and coil 2 to analyze the relationship between the deviation of the peak position of  and the current ratio of the two coils, and keep the current frequency of coil 1 at 1.05KHz and the current frequency of coil 2 at 1.00KHz. The specific method is: to keep the current frequency of coil 1 at 1.05KHz and the current frequency of coil 2 at 1.00KHz, fix the sum of the currents of the two coils to 4000A, set the current ratio of the control group to 1:1, that is, both coils are connected to 2000A ; set the current ratio of the experimental group to 1:3, that is, the coil 1 is connected to 1000A, and the coil 2 is connected to 3000A for simulation analysis.

Same as Section 3.1, draw a plane parallel to the coil at 5mm below the upper surface of the normal saline model, that is, a distance of 10mm from the lower surface of the coil, and draw the x component of the induced electric field. In Fig. 6A, the line connecting the centers of coils 1 and 2 is first defined as the y-axis, and the positive direction points to coil 2. Secondly, the midpoint of the line connecting the centers of the two coils is selected as the origin of the coordinates. Finally, the x-axis is defined as a straight line passing through the origin and perpendicular to the y-axis. The direction points to coils 3, and 4 directions. On the x-axis (x=0), select 10 points on the left and right sides with the origin as the center, and the step length is 2mm, which is called array 1 (Figure 6B); similarly, move array 1 positively along the x-axis by 5mm, which is Array 2 (Fig. 6C). `

Figure 6. The simulation results of the movement of the focus area (A) Schematic diagram of coil placement (B) Array 1 point line graph vs. line graph, the broken line marked with black triangles indicates 1:3, and the red circle marked line indicates 1:1 (C ) Array 2-point line diagram vs. line diagram, the broken line marked with black triangles indicates 1:3, and the line marked with red circles indicates 1:1 (D) The induced electric field distribution at point a at the ratio of 1:1 and 1:3 (E) Induced electric field distribution at two points c and d at a ratio of 1:3 (F) Distribution of an induced electric field at two points c and e at a ratio of 1:3.

Figure 6B shows that the dot plot lines of data 1 and 2 are symmetrical, and the peak of the induced electric field  is located at the center of the symmetry and gradually decays along the left and right, which is similar to the distribution of the induced electric field in the figure-eight focus area. In addition, the symmetry centers of  of the two data show that there is no coincidence, and the distance is 8mm (the peak is at the origin of the coordinates in the case of 1:1, and the peak is at the coordinate (-8) in the case of 1:3). The results show that changing the current ratio will affect the peak position of  , and the peak position shifts to the side with smaller current, which supports the formula (2). Excluding chance, the simulation of array 2 was carried out, and the result was similar to that of array 1. As shown in Fig 6C, the peak position of  shifted by 4mm. However, the experimental group's overall induced electric field intensity in the two arrays in Figures 6B and 6C is smaller than that of the control group, which this study will describe in the discussion in Section 4.

To further observe the overall situation of the low-frequency induced electric field envelope, select 5 points on the y-axis, namely a (0, 0), b (0, -4), c (0, -6), d (0, -2) and e(0,6). Fig 6D shows the envelope of point an at the ratio of 1:1 and 1:3, where the blue line represents 1:1. The green line represents 1:3. The results show that the envelope of the blue line is obviously better than that of the green line. There is a certain deviation of the origin when the ratio is 1:3. Both Figure 6E and Figure 6F are compared at a ratio of 1:3. In Figure 6E, the blue line represents c, and the green line represents d; in Figure 6F, the blue line represents c, and the green line represents e. Through comparison, it is found that point c and point d envelope-induced electric field is similar, but the difference between point c and point e is relatively large. The reason is that when the current ratio changes from 1:1 to 1:3, the center of symmetry (that is, the peak value of ) shifts, resulting in the focus area moving towards the negative direction of the y-axis, so the envelope electric field waveform at point e is significantly worse than that at point c.

In summary, the peak shift of  conforms to formula (2), that is, the overall focus area moves toward the side with a weak current. The reason is: Assume that the distribution of the two coils on the y-axis is consistent, that is, keep the sum of at any point on the y-axis unchanged (this is based on the symmetry of the centers of coils 1 and 2 concerning the x-axis), so the peak of  appears at the maximum value of formula (2), that is, when . Based on the above theory, when coil 2 is fed with a current of 3000A, which is greater than the 1000A current of coil 1, the induced electric field strength  near coil 2 is greater than the induced electric field strength  near coil 1, so the point of  will be biased and move to the vicinity of coil 1, the side where the current is weak.

Modified version [419-]:

Same as Section 3.1, draw a plane parallel to the coil at 20mm below the upper surface of the normal saline model, that is, a distance of 25mm from the lower surface of the coil, and draw the x component of the induced electric field. In Fig. 7A, the line connecting the centers of coils 1 and 2 is first defined as the y-axis, and the positive direction points to coil 2. Secondly, the midpoint of the line connecting the centers of the two coils is selected as the origin of the coordinates. Finally, the x-axis is defined as a straight line passing through the origin and perpendicular to the y-axis. The direction points to coils 3, and 4 directions. On the x-axis (x=0), select 15 points on the left and right sides with the origin as the center, and the step length is 5mm, which is called Array 1 (Figure 7B); similarly, move Array 1 positively along the x-axis by 5mm, which is Array 2 (Figure. 7C).

Figure 7. The simulation results of the movement of the focus area (A) Schematic diagram of coil placement (B) Array 1: the dotted line diagram at x=0, represented by a black line marked with a box for 1:1, and a red line marked with a circle for 1:5 (C ) Array 2: the dotted line diagram at x=5, represented by a black line marked with a box for 1:1, and a red line marked with a circle for 1:5 (D) The induced electric field distribution at point a at the ratio of 1:1 and 1:5,when x=0 (E) Induced electric field distribution at points b at a ratio of 1:5, when x=0.

Figure 7B shows that the dot plot lines of data 1 and 2 are symmetrical, and the peak of the induced electric field  is located at the center of the symmetry and gradually decays along the left and right, which is similar to the distribution of the induced electric field in the figure-eight focus area. In addition, the symmetry centers of  of the two data show that there is no coincidence, and the distance is 45mm (the peak is at the origin of the coordinates in the case of 1:1, and the peak is at the coordinate (45mm) in the case of 1:5). The results show that changing the current ratio will affect the peak position of , and the peak position shifts to the side with smaller current, which supports the formula (2). Excluding chance, the simulation of array 2 was carried out, and the result was similar to that of array 1. As shown in Fig 7C, the peak position of  shifted by 45mm. However, the Group 2's overall induced electric field intensity in the two arrays in Figures 7B and 7C is smaller than that of the Group 1, which this study will describe in the discussion in Section 4.

To further observe the overall situation of the low-frequency induced electric field envelope, select 2 points on the y-axis, namely a (0, 0), b (0, 45). Fig 7D shows the envelope of point a at the ratio of 1:1 and 1:5, where the black line represents 1:1. The green line represents 1:5. The results show that the envelope of the black line is obviously better than that of the green line. There is a certain deviation of the origin when the ratio is 1:5. Fig 7E shows the envelope of point b at the ratio of 1:1 and 1:5, where the gray line represents 1:1. The red line represents 1:5. The results show that the envelope of the red line is obviously better than that of the gray line. This is because in the 1:5 situation, the focus area shifted.

Modified version [545-] :

The simulation model was built with physiological saline to analyze the relationship between the change of the focus area of temporal interference magnetic stimulation and the current ratio. Figure 7B shows that when the ratio is changed from 1:1 to 1:5, the focus area is shifted from the center of the coil, that is, the origin to 45mm, and there is still symmetry, indicating that the current adjustment ratio can be shifted from the peak position of , thereby changing the position of the overall focus area.The offset of the region is consistent with the magnitude mentioned in the literature [12], both at the centimeter level. From Figures 7B and C, this study also found that the overall induced electric field modulus of the Group 2 (1:5) in array 1 and array 2 is smaller than that of the Group 1 (1:1) because the simulation in Section 3.3 is based on the sum of the two coils maintaining. Therefore, when the ratio is changed, the focus area tends to move towards the direction of the low induced electric field intensity, resulting in the overall intensity being smaller than that of the Group 1. In summary, this paper simulates the relationship between the movement of the focus area of temporal interference magnetic stimulation and the current ratio and selects two coils with different frequencies for verification and analysis. The author believes that the key to fine-tuning the focus lies in the ratio of the induced electric field of different frequencies in the focus area, rather than the number of activated coils, so this paper does not conduct simulation with multiple coils (the combination of multiple coils at two frequencies is essentially a superposition of fields).

Thank you for your question about number numbering. We have made the following modifications:

Original graph:

Figure 4A                               Figure 4G

Figure 5A   

Modified graph:

Figure 4A                                   Figure 4G

Figure 6A 

  1. The validity of the authors’ assumption that the activation occurs when the  direction of the induced electric field is parallel to ‘the axis of the nerve cells’ (meaning undefined) is an oversimplification. The authors justify this assumption with the statement  ‘according to the literature (18)’.  Referecne 18 refers to Roth and Basser’s model of the effect of stimulation a nerve fibre.  Empirical evidence from a concurrent TMS and PET study (Krieg et al2015  Neural Eng. 12 046014 DOI 10.1088/1741-2560/12/4/046014 ) reveals that brain regions activated by the TMS have significantly higher E-field components parallel to pyramidal neurons in the dendrite-to-axon orientation) and in the tangential direction (i.e., parallel to interneurons) at high gradient.

Thank you for raising the issue of our inappropriate description of literature [18]. We have made the following modifications:

Original [157-]

According to the literature [18], when the direction of the induced electric field is parallel to the axis of the nerve cells, the nerve cells are easily activated. This article assumes that the axis of the target nerve cell is parallel to the x component of the induced electric field (that is, the tangential direction of coil 1, coil 2, and coil 1, coil 5), so it is only necessary to calculate the of the x component of the induced electric field at any position n. Therefore, the above equation is corrected as (2)

Modified [200-]

According to the TMS and PET research experience provided in literature [34] and literature [35], brain regions activated by the TMS have significantly higher E-field components parallel to pyramidal neurons in the dendrite-to-axon orientation) and in the tangential direction (i.e., parallel to interneurons) at high gradient.This article assumes that the axis of the target nerve cell is parallel to the x component of the induced electric field (that is, the tangential direction of coil 1, coil 2, and coil 1, coil 5), so it is only necessary to calculate the  of the x component of the induced electric field at any position n. Therefore, the above equation is corrected as (2)

  1. The authors report simulations using two models: a saline cuboid and a five layered sphere model, with layers representing gray matter, white matter, cerebrospinal fluid, skull, and the scalp.  The sphere model is not described in adequate detail (e.g. the thickness of the layers is not specified)  

Thank you for your question about the unclear layered description of the spherical model. We have made relevant supplements, and we have also made more supplements for the source of electrical characteristic data:

Original [186-]:

the second is a temporal interference multi-target magnetic stimulation sphere model, which has five layers, including gray matter, white matter, cerebrospinal fluid, skull, and the scalp can be used to simulate the induced electric field distribution in the human brain. (2) Assign material properties. The conductivity of the saline model is set to 0.333S/m, and the relative permittivity is 100; the conductivity of the white matter is set to 0.062 S/m, and the relative permittivity is 69811; the conductivity of the gray matter is set to 0.098 S/m, relative permittivity is 69811; cerebrospinal fluid conductivity is set to 2 S/m, relative permittivity is 109; skull conductivity is set to 0.021 S/m, relative permittivity is 2702; scalp conductivity is set to 0.307 S/m, the relative permittivity is 149050; the conductivity of the array coils is set to 5.998×107S/m, and the relative permittivity is 1 [19]. (3) Add physical field-electromagnetic field to solve Maxwell's equations numerically. (4) Use the transient researcher to solve the time-domain distribution of the induced electric field intensity   of the x component, and calculate the low-frequency envelope induced electric field intensity  and modulation intensity  at any position n.

Modified version [240-]:

The other is a temporal interference multi-target magnetic stimulation sphere model, which has five layers, including scalp, skull, cerebrospinal fluid (csf), and gray matter can be used to simulate the induced electric field distribution in the human brain. The scalp, skull, cerebrospinal fluid (csf), and gray matter were 4.9mm, 7.4mm, 2.5mm, and 7.2mm, respectively. The innermost white matter of the brain was a sphere with a radius of 70mm[39-40]. Here, it should be emphasized that the reason for choosing the five layer sphere model instead of the real human brain model in this article is as follows: the concentric sphere model has been used in multiple literature[9,14,31,41], with high accuracy and little difference from the real head model. It can be used for qualitative and quantitative analysis to a certain extent; Secondly, using a real brain model, such as MRI segmentation of images, will generate a large number of units. Although many articles on coil optimization design and analysis of induced electric field distribution use real brain models for simulation, transient simulations with small steps such as time interference (usually within 0.1ms) consume more computer resources.(2) Assign material properties. The conductivity of the saline model is set to 0.333S/m, and the relative permittivity is 100; the conductivity of the white matter is set to 0.062 S/m, and the relative permittivity is 69811; the conductivity of the gray matter is set to 0.098 S/m, relative permittivity is 69811; cerebrospinal fluid conductivity is set to 2 S/m, relative permittivity is 109; skull conductivity is set to 0.021 S/m, relative permittivity is 2702; scalp conductivity is set to 0.307 S/m, the relative permittivity is 149050; the conductivity of the array coils is set to 5.998×107 S/m, and the relative permittivity is 1. The electrical parameters of human tissue used in this paper are provided by the online database of IFAC in the United States [42]. This database provides data on the electrical properties of different tissues under the influence of electromagnetic fields from 10 Hz to 100 GHz, which is more targeted than other conductivity data [43].

  1. My greatest misgiving is the limited clinical relevance of the effects reported.  Despite the fact that a major reason for employing temporal interference magnetic stimulation is to achieve stimulation deep within the brain, the authors report the effects of stimulation only at shallow depths. In the saline cuboid model, the effects of stimulation are reported in a plane parallel to the surface and 5 mm below the surface.  For the sphere model, the target plane is described as  10 mm below the coil which is located 5 mm above the highest point of the surface sphere. This implies that target plane is 5mm or less below the surface of the sphere. The authors do not say what tissue type is located at this depth.  One would hope it would be gray matter but if so, the combined thickness of scalp, bone and CSF space is not realistic.

Thank you for your question about the selection of the depth of stimulation in this article. After careful consideration, we agree with you very much, and apologize to you again for the unreasonableness of our scheme. For this, we re planned the simulation scheme, extending the stimulation depth from the previous 5mm to 20mm, which is enough to stimulate the gray matter area. The article is changed as follows, and finally we provided the depth distribution map of modulation intensity:

Original [214-]:

3.1. Multi-target temporal interference magnetic stimulation in a saline model

Select the upper surface of the normal saline to be 5mm, that is, the distance from the lower surface of the coil is 10mm, draw the induced electric field of the x component as a plane map, and select the moment when the modulus is maximum. The position of the first group of simulation experiment coils is shown in Fig. 4A and Fig. 4B. It is stipulated that the center of the central coil is the coordinate origin, the line connecting the centers of coils 1 and 2 is the y-axis, the positive direction points to coil 2, the x-axis is a straight line perpendicular to the y axis, and the positive direction points to coils 3 and 4. Figure 4C has two focal areas located between coils 1 and 2 and coils 1 and 5, respectively, which meet the needs of multi-target stimulation. In addition, there are two areas with lower peak values. The reasons for this will be discussed in Section 4 of this article. Define area 1 to be located between coils 1 and 2, and area 2 to be located between coils 1 and 5. Select 4 positions (①②③④) in area 1 and area 2, respectively, to investigate the distribution of the induced electric field of the x component. The positions are (0, 27), (35, 27), (0, 50), (0, -27). The green line in Figure 4D indicates point ① in area 1, which is located at the center of coils 1 and 2. At this time, , so the modulation effect is the best, and the  is 11.42V/m; (D) The middle blue line represents point ②, which is located 35mm to the right of point ①, that is, the positive direction of the x-axis. Although it is far away from the center, it is at the same distance from the two coils, and still satisfies the condition of, but compared with point ①, the sum of is small, so the will have a certain attenuation, and its value is 9.45V/m; the black line in Figure 4E represents point ③, Located at 230mm above ①, it is not only far away from the center of the area, but also has an unequal distance from the two coils, resulting in an excessively large difference of  at this point, which in turn affects the envelope of the entire waveform. Its value is 3.54V/m. Similarly, considering the symmetry, the distribution trend of  located in the second focal area is similar to that of the first area, as shown in the red curve of point ④ in Figure 4F, and its modulation intensity  is 11.48V/m. The arrangement of coils 1, 5, and 6 in the second group of simulations are shown in Figure 4H. The figure shows the distribution of the induced electric field of the x component, and the moment with the maximum modulus is also selected for analysis. The results show that there are two high-peak focusing areas in the figure, which are located between coil 1 and coil 5 and coil 1 and coil 6, respectively. In addition, two low-peak focus areas are caused by the same reasons as the first group, which will be analyzed in Section 4. The reason for the two high-peak focusing areas is the same as the principle of the figure-eight coil, so this research will not go into details here, and focus on the three points selected in Figures 4H and 4I, namely points ⑤ (0, -26), ⑥ (-20, -16), ⑦ (-28, -46). Point ⑤ (the blue line in Figure 4I) is located at the junction of coil 1 and coil 5, and the  is relatively high (12.03V/m); point ⑥ is located at the junction of coil 1 and coil 6. Electric field modulus (red line in Figure 4I), so the overall induced electric field intensity is small, but still maintains a high modulation intensity (9.32V/m); point ⑦ is located at the junction of coil 5 and coil 6, theoretically Since the currents passed into the two coils are high frequency, the overall envelope effect is poor, and the modulation intensity is only (1.78V/m). However, it is found that there is still a certain modulation depth at this point, presumably due to the interference of coil 1. In summary, although there is a modulation envelope at point ⑦, that is, the junction of coils 5 and 6, it can be ignored because the distance between coil 1 and this area is relatively small, and the interference components are small, which can be eliminated by optimizing the coil layout in the future; in addition, the coils 5 and 6 are supplied with counterclockwise currents, causing the direction of the induced electric field in the junction area to be reversed, eliminating most of the induced electric field strength to a certain extent.

Figure 4. Double-target simulation results (A), (B) Schematic diagram of the arrangement of coils 1, 2, and 5 in the saline model (C) Simulation results of 1, 2, and 5 focus areas (D) ①, ② Time-domain distribution of induced electric field (E) Time-domain distribution of induced electric field at points ① and ③ (F) Time-domain distribution of induced electric field at point 4 (G) Schematic diagram of the arrangement of coils 1, 5 and 6 in the saline model (H) 1 , 5, 6 focus area simulation results (I) Time-domain distribution of induced electric field at points ⑤, ⑥, ⑦.

3.2. Multi-target temporal interference magnetic stimulation in Spherical model

Select a five-layer sphere model to establish a simulation model of human brain multi-target temporal interference magnetic stimulation. The array coils is located 5mm above the highest point on the sphere's surface, and the rest of the settings are the same as those of normal saline. Saline is consistent. Select 10mm of the lower surface of the coil as a plane parallel to the coil, which is the 5mm depth of the sphere, and draw the distribution of the induced electric field of the x component. The results showed that there were two focal regions in Figure 5A、5B, and the positions were similar to the focal region distribution of the saline model. Select points ⑧, ⑨, and ⑩ to analyze the , and the positions are (0, 30), (35, 27), (0, -45), respectively. The yellow, red, and blue lines in Figure 5C represent the induced electric field time-domain variation curves of points ⑧, ⑨, and ⑩, respectively. The results show that point ⑧ is located at the center of coils 1 and 2, so the is the largest; point ⑨ along the y axis origin shifts 8mm to the positive x direction, and the  shows attenuation; point  is located outside the focus area, and the  is the smallest, almost none.

Figure 5. Sphere model simulation results (A) Schematic diagram of the arrangement of coils 1, 2, 5 under the sphere head model (B) Simulation of two focus areas in the sphere head model Results (C) Temporal distribution of induced electric field at points ⑧, ⑨, ⑩.

Modified version [301-]:

3.1. Multi-target temporal interference magnetic stimulation in a saline model

Select the upper surface of the normal saline to be 20mm, that is, the distance from the lower surface of the coil is 25mm, draw the induced electric field of the x component as a plane map, and select the moment when the modulus is maximum. The position of the first group of simulation experiment coils is shown in Fig. 4A and Fig. 4B. It is stipulated that the center of the central coil is the coordinate origin, the line connecting the centers of coils 1 and 2 is the y-axis, the positive direction points to coil 2, the x-axis is a straight line perpendicular to the y axis, and the positive direction points to coils 3 and 4. Figure 4C has two focal areas located between coils 1 and 2 and coils 1 and 5, respectively, which meet the needs of multi-target stimulation. In addition, there are two weaker focusing regions located on the outer edges of coil 5 and coil 2, respectively. The reason for its occurrence is related to the current flowing through the outer edge of the two coils. At the same time, this article selects the direction of the induced electric field as the x direction, so the focus area is concentrated near the y-axis. Despite the presence of these two weakly focused regions, coils 2 and 5 are connected with continuous high-frequency stimulation signals, so there is no cortical response in this region.Define area 1 to be located between coils 1 and 2, and area 2 to be located between coils 1 and 5. Select 4 positions (①②③④) in area 1 and area 2, respectively, to investigate the distribution of the induced electric field of the x component. The positions are (0, 26), (37, 26), (0, 50), (0, -26).

The green line in Figure 4D indicates point ① in area 1, which is located at the center of coils 1 and 2. At this time, , so the modulation effect is the best, and the  is 4.5V/m; (D) The middle blue line represents point ②, which is located 37mm to the right of point ①, that is, the positive direction of the x-axis. Although it is far away from the center, it is at the same distance from the two coils, and still satisfies the condition of , but compared with point ①, the sum of  is small, so the  will have a certain attenuation, and its value is 3.5V/m; the black line in Figure 4E represents point ③, Located at 23mm above ①, it is not only far away from the center of the area 1, but also has an unequal distance from the two coils, resulting in an excessively large difference of  at this point, which in turn affects the envelope of the entire waveform. Its value is 1.1V/m. Similarly, considering the symmetry, the distribution trend of  located in the second focal area is similar to that of the area 1, as shown in the red line of point ④ in Figure 4F, and its amplitude modulation intensity  is approximate 4.5V/m. The arrangement of coils 1, 5, and 6 in the second group of simulations are shown in Figure 4G. The Figure 4H shows the distribution of the induced electric field of the x component, and the moment with the maximum modulus is also selected for analysis. The results show that there are two high-peak focusing areas in the figure, which are located between coil 1 and coil 5 and coil 1 and coil 6, respectively. In addition, two low-peak focus areas are caused by the same reasons as the first group, which will be analyzed in Section 4. The reason for the two high-peak focusing areas is the same as the principle of the figure-eight coil, so this research will not go into details here, and focus on the three points selected in Figures 4H, namely points ⑤ (0, -26), ⑥ (-20, -16), ⑦ (-28, -46). Point ⑤ (the blue line in Figure 4I) is located at the junction of coil 1 and coil 5, and the  is relatively high (5.6V/m); point ⑥ is located at the junction of coil 1 and coil 6. Electric field modulus (red line in Figure 4I), so the overall induced electric field intensity is small, but still maintains a high amplitude modulation intensity  (5.1V/m); point ⑦ is located at the junction of coil 5 and coil 6, theoretically since the currents passed into the two coils are same high frequency, the overall envelope effect is poor, and the amplitude modulation intensity is only (1.4V/m). However, it is found that there is still a certain amplitude modulation intensity at this point, presumably due to the interference of coil 1. In summary, although there is a modulation envelope at point ⑦, that is, the junction of coils 5 and 6, it can be ignored because the distance between coil 1 and this area is relatively small, and the interference components are small, which can be eliminated by optimizing the coil layout in the future; in addition, the coils 5 and 6 are supplied with counterclockwise currents, causing the direction of the induced electric field in the junction area to be reversed, eliminating most of the induced electric field strength to a certain extent.

Figure 4. (A) Multi-target simulation results, all arrows indicate current direction (B) Schematic diagram of the arrangement of coils 1, 2, and 5 in the saline model (C) Simulation results of 1, 2, and 5 focus areas (D) ①, ② Time-domain distribution of induced electric field (E) Time-domain distribution of induced electric field at points ① and ③ (F) Time-domain distribution of induced electric field at point 4 (G) Schematic diagram of the arrangement of coils 1, 5 and 6 in the saline model (H) 1, 5, 6 focus area simulation results (I) Time-domain distribution of induced electric field at points ⑤, ⑥, ⑦.ouble-target simulation results (A), (B) Schematic diagram of the arrangement of coils 1, 2, and 5 in the saline model (C) Simulation results of 1, 2, and 5 focus areas (D) ①, ② Time-domain distribution of induced electric field (E) Time-domain distribution of induced electric field at points ① and ③ (F) Time-domain distribution of induced electric field at point 4 (G) Schematic diagram of the arrangement of coils 1, 5 and 6 in the saline model (H) 1 , 5, 6 focus area simulation results (I) Time-domain distribution of induced electric field at points ⑤, ⑥, ⑦.(Revise(1))

For the depth distribution of induced electric field, the z-direction segmentation is carried out along the y-axis in Figure 4A to obtain the depth induced electric field distribution of coils 1, 2 and 5 , in physiological saline. The results are shown in Figure 5A and Figure 5B. Figure 5A shows the induced electric field distribution no temporal interference, while Figure 5B shows the  distribution under temporal interference. By comparison, it was found that as the depth increased, the focusing area under temporal interference was significantly better than that of the group without time interference. Although the focus area is mainly concentrated at a depth of 10mm, it may not reach the gray area. The reason is that this simulation uses a qualitative simulation analysis at a frequency of 1KHz. To stimulate deeper areas, researchers can increase the frequency of the differential signal to the level of 10KHz, as used in reference [14].

Figure 5 Depth distribution map of induced electric field (A) Distribution of deep induction electric field no temporal interference. (B) Distribution of  temporal interference

  1. In the case of the manipulation of the relative current in a pair of coils, changing the current ratio from 1:1 to 1:3 produces a  horizontal shift of the peak amplitude modulation that is small compared with the half width of the peak, implying a shift in peak predicted neural excitation is  that is less than the spatial resolution of the stimulation profile. As far as I can see, the magnitude of the reported effects of interest are of minimal clinical relevance.  The implication that their findings indicate that multi- target temporal interference magnetic stimulation with array coils can simultaneously stimulate multiple network nodes in the brain region in a clinically meaningful manner is not justified.

Thank you for your suggestion regarding the short moving distance of the focus area (only 8mm) in the 1:3 ratio. In response to this issue, we have adjusted the simulation scheme to 1:5 and made the following modifications. In a ratio of 1:5, the focus area was moved as much as possible by 45mm, sufficient to stimulate different associated nodes.

Original [302-]:

In the normal saline simulation model, select coil 1 and coil 2 to analyze the relationship between the deviation of the peak position of  and the current ratio of the two coils, and keep the current frequency of coil 1 at 1.05KHz and the current frequency of coil 2 at 1.00KHz. The specific method is: to keep the current frequency of coil 1 at 1.05KHz and the current frequency of coil 2 at 1.00KHz, fix the sum of the currents of the two coils to 4000A, set the current ratio of the control group to 1:1, that is, both coils are connected to 2000A ; set the current ratio of the experimental group to 1:3, that is, the coil 1 is connected to 1000A, and the coil 2 is connected to 3000A for simulation analysis.

Same as Section 3.1, draw a plane parallel to the coil at 5mm below the upper surface of the normal saline model, that is, a distance of 10mm from the lower surface of the coil, and draw the x component of the induced electric field. In Fig. 6A, the line connecting the centers of coils 1 and 2 is first defined as the y-axis, and the positive direction points to coil 2. Secondly, the midpoint of the line connecting the centers of the two coils is selected as the origin of the coordinates. Finally, the x-axis is defined as a straight line passing through the origin and perpendicular to the y-axis. The direction points to coils 3, and 4 directions. On the x-axis (x=0), select 10 points on the left and right sides with the origin as the center, and the step length is 2mm, which is called array 1 (Figure 6B); similarly, move array 1 positively along the x-axis by 5mm, which is Array 2 (Fig. 6C). `

Figure 6. The simulation results of the movement of the focus area (A) Schematic diagram of coil placement (B) Array 1 point line graph vs. line graph, the broken line marked with black triangles indicates 1:3, and the red circle marked line indicates 1:1 (C ) Array 2-point line diagram vs. line diagram, the broken line marked with black triangles indicates 1:3, and the line marked with red circles indicates 1:1 (D) The induced electric field distribution at point a at the ratio of 1:1 and 1:3 (E) Induced electric field distribution at two points c and d at a ratio of 1:3 (F) Distribution of an induced electric field at two points c and e at a ratio of 1:3.

Figure 6B shows that the dot plot lines of data 1 and 2 are symmetrical, and the peak of the induced electric field  is located at the center of the symmetry and gradually decays along the left and right, which is similar to the distribution of the induced electric field in the figure-eight focus area. In addition, the symmetry centers of  of the two data show that there is no coincidence, and the distance is 8mm (the peak is at the origin of the coordinates in the case of 1:1, and the peak is at the coordinate (-8) in the case of 1:3). The results show that changing the current ratio will affect the peak position of  , and the peak position shifts to the side with smaller current, which supports the formula (2). Excluding chance, the simulation of array 2 was carried out, and the result was similar to that of array 1. As shown in Fig 6C, the peak position of  shifted by 4mm. However, the experimental group's overall induced electric field intensity in the two arrays in Figures 6B and 6C is smaller than that of the control group, which this study will describe in the discussion in Section 4.

To further observe the overall situation of the low-frequency induced electric field envelope, select 5 points on the y-axis, namely a (0, 0), b (0, -4), c (0, -6), d (0, -2) and e(0,6). Fig 6D shows the envelope of point an at the ratio of 1:1 and 1:3, where the blue line represents 1:1. The green line represents 1:3. The results show that the envelope of the blue line is obviously better than that of the green line. There is a certain deviation of the origin when the ratio is 1:3. Both Figure 6E and Figure 6F are compared at a ratio of 1:3. In Figure 6E, the blue line represents c, and the green line represents d; in Figure 6F, the blue line represents c, and the green line represents e. Through comparison, it is found that point c and point d envelope-induced electric field is similar, but the difference between point c and point e is relatively large. The reason is that when the current ratio changes from 1:1 to 1:3, the center of symmetry (that is, the peak value of ) shifts, resulting in the focus area moving towards the negative direction of the y-axis, so the envelope electric field waveform at point e is significantly worse than that at point c.

In summary, the peak shift of  conforms to formula (2), that is, the overall focus area moves toward the side with a weak current. The reason is: Assume that the distribution of the two coils on the y-axis is consistent, that is, keep the sum of at any point on the y-axis unchanged (this is based on the symmetry of the centers of coils 1 and 2 concerning the x-axis), so the peak of  appears at the maximum value of formula (2), that is, when . Based on the above theory, when coil 2 is fed with a current of 3000A, which is greater than the 1000A current of coil 1, the induced electric field strength  near coil 2 is greater than the induced electric field strength  near coil 1, so the point of  will be biased and move to the vicinity of coil 1, the side where the current is weak.

Original [412-]:

The simulation model was built with physiological saline to analyze the relationship between the change of the focus area of temporal interference magnetic stimulation and the current ratio. Figure 6B shows that when the ratio is changed from 1:1 to 1:3, the focus area is shifted from the center of the coil, that is, the origin to 8mm, and there is still symmetry, indicating that the current adjustment ratio can be shifted from the peak position of  , thereby changing the position of the overall focus area. In addition, it can be observed that on the y-axis of x=5, array 2 is selected to perform simulation analysis again, and it is found that the peak offsets of the two data  are both smaller than x=0. The author believes that the reason is that the coordinate points of the two data in array 2 are far away from the centers of the two coils, so the amplitude of the induced electric field  is relatively small overall. But in general, data 2 of the two arrays has  peak shift, and the overall focus area has moved. From Figures 6B and C, this study also found that the overall induced electric field modulus of the experimental group (1:3) in array 1 and array 2 is smaller than that of the control group (1:1) because the simulation experiment in Section 3.3 is based on the sum of the two coils maintaining. Therefore, when the ratio is changed, the focus area tends to move towards the direction of the low induced electric field intensity, resulting in the overall intensity being smaller than that of the control group. In summary, this paper simulates the relationship between the movement of the focus area of temporal interference magnetic stimulation and the current ratio and selects two coils with different frequencies for verification and analysis. The author believes that the key to fine-tuning the focus lies in the ratio of the induced electric field of different frequencies in the focus area, rather than the number of activated coils, so this paper does not conduct experiments with multiple coils (the combination of multiple coils at two frequencies is essentially a superposition of fields).

Modified version [291-]:

In the normal saline simulation model, select coil 1 and coil 2 to analyze the relationship between the deviation of the peak position of  and the current ratio of the two coils, and keep the current frequency of coil 1 at 1.00KHz and the current frequency of coil 2 at 1.05KHz. The specific method is: to keep the current frequency of coil 1 at 1.00KHz and the current frequency of coil 2 at 1.05KHz, fix the sum of the currents of the two coils to 4000A, set the current ratio of the Group 1 to 1:1, that is, both coils are connected to 2000A ; set the current ratio of the Group 2 to 1:5, that is, the coil 1 is connected to 3333A, and the coil 2 is connected to 667A for simulation analysis.

Modified version [419-]:

Same as Section 3.1, draw a plane parallel to the coil at 20mm below the upper surface of the normal saline model, that is, a distance of 25mm from the lower surface of the coil, and draw the x component of the induced electric field. In Fig. 7A, the line connecting the centers of coils 1 and 2 is first defined as the y-axis, and the positive direction points to coil 2. Secondly, the midpoint of the line connecting the centers of the two coils is selected as the origin of the coordinates. Finally, the x-axis is defined as a straight line passing through the origin and perpendicular to the y-axis. The direction points to coils 3, and 4 directions. On the x-axis (x=0), select 15 points on the left and right sides with the origin as the center, and the step length is 5mm, which is called Array 1 (Figure 7B); similarly, move Array 1 positively along the x-axis by 5mm, which is Array 2 (Figure. 7C).

Figure 7. The simulation results of the movement of the focus area (A) Schematic diagram of coil placement (B) Array 1: the dotted line diagram at x=0, represented by a black line marked with a box for 1:1, and a red line marked with a circle for 1:5 (C ) Array 2: the dotted line diagram at x=5, represented by a black line marked with a box for 1:1, and a red line marked with a circle for 1:5 (D) The induced electric field distribution at point a at the ratio of 1:1 and 1:5,when x=0 (E) Induced electric field distribution at points b at a ratio of 1:5, when x=0.

Figure 7B shows that the dot plot lines of data 1 and 2 are symmetrical, and the peak of the induced electric field  is located at the center of the symmetry and gradually decays along the left and right, which is similar to the distribution of the induced electric field in the figure-eight focus area. In addition, the symmetry centers of  of the two data show that there is no coincidence, and the distance is 45mm (the peak is at the origin of the coordinates in the case of 1:1, and the peak is at the coordinate (45mm) in the case of 1:5). The results show that changing the current ratio will affect the peak position of , and the peak position shifts to the side with smaller current, which supports the formula (2). Excluding chance, the simulation of array 2 was carried out, and the result was similar to that of array 1. As shown in Fig 7C, the peak position of  shifted by 45mm. However, the Group 2's overall induced electric field intensity in the two arrays in Figures 7B and 7C is smaller than that of the Group 1, which this study will describe in the discussion in Section 4.

To further observe the overall situation of the low-frequency induced electric field envelope, select 2 points on the y-axis, namely a (0, 0), b (0, 45). Fig 7D shows the envelope of point a at the ratio of 1:1 and 1:5, where the black line represents 1:1. The green line represents 1:5. The results show that the envelope of the black line is obviously better than that of the green line. There is a certain deviation of the origin when the ratio is 1:5. Fig 7E shows the envelope of point b at the ratio of 1:1 and 1:5, where the gray line represents 1:1. The red line represents 1:5. The results show that the envelope of the red line is obviously better than that of the gray line. This is because in the 1:5 situation, the focus area shifted.

Modified version [545-] :

The simulation model was built with physiological saline to analyze the relationship between the change of the focus area of temporal interference magnetic stimulation and the current ratio. Figure 7B shows that when the ratio is changed from 1:1 to 1:5, the focus area is shifted from the center of the coil, that is, the origin to 45mm, and there is still symmetry, indicating that the current adjustment ratio can be shifted from the peak position of , thereby changing the position of the overall focus area.The offset of the region is consistent with the magnitude mentioned in the literature [12], both at the centimeter level. From Figures 7B and C, this study also found that the overall induced electric field modulus of the Group 2 (1:5) in array 1 and array 2 is smaller than that of the Group 1 (1:1) because the simulation in Section 3.3 is based on the sum of the two coils maintaining. Therefore, when the ratio is changed, the focus area tends to move towards the direction of the low induced electric field intensity, resulting in the overall intensity being smaller than that of the Group 1. In summary, this paper simulates the relationship between the movement of the focus area of temporal interference magnetic stimulation and the current ratio and selects two coils with different frequencies for verification and analysis. The author believes that the key to fine-tuning the focus lies in the ratio of the induced electric field of different frequencies in the focus area, rather than the number of activated coils, so this paper does not conduct simulation with multiple coils (the combination of multiple coils at two frequencies is essentially a superposition of fields).

Reviewer 4 Report

This paper presents an design of coils array for magnetic stimulation.
I have few remarks that must be addressed before give the recommendation for publication on these transactions.
First: the abstract mus include the most important indicators and coil dimensions and disnaces between them, as well as the number of coils in the array.
With relation to figure 2, how it is planed to implement such a setup? This is important to allow the ocasional reader to implement a magnetic stimulation system, either with this configuration of coils or with other type of configurations.
In the state-of-the-art in the introduction, it is important to reffer complementary works on neural stimulation with electric fields from Boyden et al (10.1016/j.cell.2017.05.024), optogenetic implementations by Correia et al (doi.org/10.3390/mi9090473), as well as DBS implementations by Nordi et al (10.3390/electronics11060939).
Clarify how the electrical parameters of the saline solution, skull, cerebrospinal fluid conductivity and so on were selected. If possible include references.
Which material of the cois is modelled? Copper? Alluminium? What is its thickness? What is the (insulating) boundary material that surronds the coils are intended to be used?
It is not clear how these dimensions and spacings were obtained. Please clarify.

Author Response

Dear  Reviewer :

Thanks very much for your kind letter about the review of our paper titled “Optimal Design of Array Coils for Multi-target Adjustable Electromagnetic Brain Stimulation System” (ID: bioengineering-2314035). for publication in “bioengineering”. We have finished the proof reading and checking carefully, and some corrections about the answers to the queries are provided below.The details of the modification are in the attachment.

                                                     Author:Dr. Tingyu Wang、Mr. Lele Yan、Dr. Xinsheng Yang、Prof. Duyan Geng、Prof. Dr. Guizhi Xu *、Prof. Alan Wang *

  1. The abstract must include the most important indicators and coil dimensions and disances between them, as well as the number of coils in the array.

Thank you very much for raising the issue of introducing some important parameters, coil quantity, size, and distance in detail in the abstract. We have made relevant modifications to this:

Original [14-]:

Abstract: Temporal interference magnetic stimulation is a novel noninvasive deep brain neuromodulation technology, which can solve the problem of balance between focus area and stimulation depth, and can stimulate deep targets without activating superficial non-target areas, improving the accuracy of stimulation. However, at present, the stimulation target of this technology is relatively single, and it is difficult to realize the coordinated stimulation of multiple brain regions, which limits its application in the modulation of multiple nodes in the brain network. This paper first proposes a multi-target temporal interference magnetic stimulation system with array coils, then establishes human tissue fluid and human brain sphere models, discusses the relationship between the movement of the focus area and the coil current ratio under temporal interference again, and finally verifies the multiple targets through simulation software. The realization of the target point and analysis of the law of the movement of the focus area has been presented. The conclusion is that multi-target temporal interference magnetic stimulation with array coils can simultaneously stimulate multiple network nodes in the brain region; rough positioning can be performed by controlling the conduction of different coils, fine-tuning the position by changing the current ratio of the conduction coils, and realizing accurate stimulation of multiple targets in the brain area.

Modified version [14-]:

Abstract: Temporal interference magnetic stimulation is a novel noninvasive deep brain neuromodulation technology, which can solve the problem of balance between focus area and stimulation depth. However, at present, the stimulation target of this technology is relatively single, and it is difficult to realize the coordinated stimulation of multiple brain regions, which limits its application in the modulation of multiple nodes in the brain network. This paper first proposes a multi-target temporal interference magnetic stimulation system with array coils. The array coils are composed of seven coil units with an outer radius of 25mm, and the spacing between coil units is 2mm. Secondly, a model of human tissue fluid and the human brain sphere are established. Finally, the relationship between the movement of the focus area and the amplitude ratio of the difference frequency excitation source , under time interference is discussed. The results show that in the case of a ratio of 1:5, the peak position of the amplitude modulation intensity of the induced electric field has moved 45mm, that is the movement of the focus area is related to the amplitude ratio of the difference frequency excitation source. The conclusion is that multi-target temporal interference magnetic stimulation with array coils can simultaneously stimulate multiple network nodes in the brain region; rough positioning can be performed by controlling the conduction of different coils, fine-tuning the position by changing the current ratio of the conduction coils, and realizing accurate stimulation of multiple targets in the brain area.

  1. With relation to figure 2, how it is planed to implement such a setup? This is important to allow the ocasional reader to implement a magnetic stimulation system, either with this configuration of coils or with other type of configurations

Thank you very much for raising the question about how the system in Figure 2 operates. In the original text, we provided sufficient description, and now we have revised this section:

Original [112-]:

The multi-target temporal interference system based on array coils is shown in Figure 3, which consists of a power module, DC/AC module, control module and coil array. One phase multiple power supply structure is adopted, in which capacitors and converters form a cell, which can control the stimulation mode of one coil. By cascading multiple cells, the number of coils can be expanded from 1 to n, thereby realizing multi-target temporal interference of array coils. In addition, the control terminal of this system can transmit control commands such as frequency and amplitude to the discharge module, and can also monitor the charging voltage of the DC capacitor and the stimulating current of the stimulating coil in real time to prevent overvoltage and overcurrent. In this paper, through the COMSOL simulation verification, multi-target synergistic stimulation can be realized by turning on multiple sets of coils simultaneously; changing the coils' current ratio can realize fine-tuning of the stimulation targets without moving the coil position.

Modified version [145-]:

The multi-target temporal interference system based on array coils is shown in Figure 2, which consists of a power module, DC/AC module, control module, and array coils. The high-frequency LCC resonant circuit constitutes a single-phase multi-channel charging structure, which has the advantage of constant current and fast charging[23].Capacitors and converters form a cell, which can control the stimulation mode of one coil. By cascading multiple cells, the number of coils can be expanded from 1 to n, thereby realizing multi-target temporal interference of array coils. The control terminal of this system can select the opening of a certain coil unit. In addition, it can transmit control commands such as frequency and amplitude to the discharge module, and can also monitor the charging voltage of the DC capacitor and the stimulating current of the stimulating coil in real time to prevent overvoltage and overcurrent[24-28]. In this paper, through the COMSOL simulation verification, multi-target synergistic stimulation can be realized by turning on multiple sets of coils simultaneously; changing the coils' current ratio can realize fine-tuning of the stimulation targets without moving the coil position.

  1. In the state-of-the-art in the introduction, it is important to reffer complementary works on neural stimulation with electric fields from Boyden et al (10.1016/j.cell.2017.05.024), optogenetic implementations by Correia et al (doi.org/10.3390/mi9090473), as well as DBS implementations by Nordi et al (10.3390/electronics11060939).

Thank you for your valuable comments. Regarding the literature citation, we have made a few revisions and added the literature you suggested to cite.

Grossman et al. conducted deep brain electrical stimulation on mice in 2017, using time interference electrical stimulation technology. Two pairs of electrodes were fed with currents at frequencies of 2000Hz and 2010Hz, with a frequency difference of 10Hz. They also applied anti-phasic current drive technology to eliminate crosstalk between the two current sources. The results showed that the 10 Hz differential frequency envelope could cause nerve discharges synchronized with the envelope. At the same time, the team successfully recorded the responses of neurons at different depths in the mouse hippocampus through patch clamp technology. In addition, the team also studied the effect of the current amplitude ratio of the two groups of electrodes on the focus area[9]

In addition, in recording neural activity and monitoring the focal area of deep brain stimulation. S. Beatriz Goncalves et al present a single LED optrode neural tool, capable of assessing tissue temperature over time, and thermal imaging of brain tissue near the stimulation focus. RTD’s average accuracy of 0.2℃ at a normal body temperature of 37℃, and take into account the ability of electrical recording and light stimulation at the same time. The RTD thin film is integrated into the silicon carbon needle to adapt to light stimulation, electrophysiological recording point, temperature sensing, etc. This electrode has great potential in recording and stimulating neural activity[16]. Tiago Matheus Nordi et al proposed a biopotential acquisition system that can be used in deep brain stimulation. The core component is a low-noise amplifier (LNA), which is designed in a voltage/low-power CMOS process, and its area is only 122μm×283μm. The test results show that the gain is 38.6 dB, the bandwidth is −3 dB, the frequency is 2.3 kHz, and the power consumption is 2.8 μW. It is expected to be applied in deep brain stimulation in the future[17].

  1. Clarify how the electrical parameters of the saline solution, skull, cerebrospinal fluid conductivity and so on were selected. If possible include references.

Thank you very much for raising the issue of insufficient description of electrical characteristics parameters such as physiological saline, skull, and scalp in the original text. We have made relevant modifications to address this issue:

Original [189-]:

(2) Assign material properties. The conductivity of the saline model is set to 0.333S/m, and the relative permittivity is 100; the conductivity of the white matter is set to 0.062 S/m, and the relative permittivity is 69811; the conductivity of the gray matter is set to 0.098 S/m, relative permittivity is 69811; cerebrospinal fluid conductivity is set to 2 S/m, relative permittivity is 109; skull conductivity is set to 0.021 S/m, relative permittivity is 2702; scalp conductivity is set to 0.307 S/m, the relative permittivity is 149050; the conductivity of the array coils is set to 5.998×107S/m, and the relative permittivity is 1 [19].

Modified version [255-]:

(2) Assign material properties. The conductivity of the saline model is set to 0.333S/m, and the relative permittivity is 100; the conductivity of the white matter is set to 0.062 S/m, and the relative permittivity is 69811; the conductivity of the gray matter is set to 0.098 S/m, relative permittivity is 69811; cerebrospinal fluid conductivity is set to 2 S/m, relative permittivity is 109; skull conductivity is set to 0.021 S/m, relative permittivity is 2702; scalp conductivity is set to 0.307 S/m, the relative permittivity is 149050; the conductivity of the array coils is set to 5.998×107 S/m, and the relative permittivity is 1. The electrical parameters of human tissue used in this paper are provided by the online database of IFAC in the United States [42]. This database provides data on the electrical properties of different tissues under the influence of electromagnetic fields from 10 Hz to 100 GHz, which is more targeted than other conductivity data [43].

  1. Which material of the cois is modelled? Copper? Alluminium? What is its thickness? What is the (insulating) boundary material that surronds the coils are intended to be used?
    It is not clear how these dimensions and spacings were obtained. Please clarify.

Thank you for raising the issue of insufficient description of coil size, material, and thickness in the original text. We have made the following modifications

Original [128-]:

All the coils are circular coils of the same size, and the arrangement is shown in Figures 3A and 3B. The center of the circular coil numbered 1 coincides with the center of the array, which is called the center coil; then the remaining numbered coils (from small to large) are arranged clockwise around the center coil. The coil radius is 25mm, the wire diameter is 2mm, the number of turns is 6, and the material is metallic copper with good conductivity.

Modified version [128-]:

All coils were circular coils of the same size, arranged as shown in Figures 3A and 3B. The center of the circular coil numbered 1 coincides with the center of the array, which is called the central coil; the remaining numbered coils (from small to large) are arranged clockwise around the central coil. The selection of radius has two factors: firstly, considering that the diameter of the five-layer sphere model is 184mm, the head circumference of adults is between 540-580mm, and there are at least three circles in the same column; Secondly, the coil radius is approximately positively correlated with the stimulus intensity. Specifically, as the coil radius increases, the cortical stimulus intensity gradually increases. So the coil unit is selected with an outer radius of 25mm [29]. The wire diameter is 2mm, the number of turns is 6, and the material is copper with good conductivity. The shell is made of an insulating epoxy resin board, which is characterized by good mechanical properties under medium temperature conditions, stable electrical properties under high humidity conditions, good heat resistance and moisture resistance, and can protect personnel safety.

Round 2

Reviewer 1 Report

   The authors explained the questions I raised and made appropriate revisions to the manuscript. I look forward to the time when this technology is used in clinical practice.

Author Response

Dear Reviewer:

Thank you for reviewing our manuscript entitled “Optimal Design of Array Coils for Multi-target Adjustable Electromagnetic Brain Stimulation System” (ID: bioengineering-2314035). Thank you for your review of this article. Your comments are rigorous and meaningful, and through your review comments, our article structure, vocabulary, and grammar have become more standardized. Finally, thank you again for your contribution to this article. Wishing you a happy life.

                                  Author:Dr. Tingyu Wang、Mr. Lele Yan、Dr. Xinsheng Yang、Prof. Duyan Geng、Prof. Dr. Guizhi Xu *、Prof. Alan Wang *

Reviewer 2 Report

The authors addressed all the raised issues.

Author Response

Dear Reviewer:

Thank you for reviewing our manuscript entitled “Optimal Design of Array Coils for Multi-target Adjustable Electromagnetic Brain Stimulation System” (ID: bioengineering-2314035). All your comments are meaningful to the article, and through your review, our essay has become more rigorous, and our grammar, vocabulary, and writing have become more standard. Thank you again for your work and have a great life.

                                  Author:Dr. Tingyu Wang、Mr. Lele Yan、Dr. Xinsheng Yang、Prof. Duyan Geng、Prof. Dr. Guizhi Xu *、Prof. Alan Wang *

Reviewer 3 Report

The revised version has been improvd substantially.  The organization of the material is better. For example, the description of the modelling of the stimulation is provided in the methods section. [lines 263-291] . The grammar is improved though further editing of language use is required.

The authors provide an improved account of the evidence that the temporal interference technique works (at least in animals) [line 58-75]. They provide additional discussion of the evidence justifying the calculation of the component of electric field parallel to the axis of the pyramidal neurons [lines 195-198]. They have improved the description of the sphere model [line 236-246]

The depth of the stimulation target has been increased from 5mm to 20 mm, which is more relevant for demonstrating the feasibility of deep stimulation.  The addition of figure 5 depicting the variation of field with increasing depth is informative.

In the revised version, the spatial resolution depicted in figure 7 is greatly improved

Minor issues:

line 48: ‘make neurons depolarization’ should be ‘make neurons depolarize’

line 102: the initial RTD should be defined at first use.

line 362; the grammar needs correction.

Author Response

Dear  Reviewer:

Thank you for reviewing our manuscript entitled “Optimal Design of Array Coils for Multi-target Adjustable Electromagnetic Brain Stimulation System” (ID: bioengineering-2314035).  The grammar, vocabulary and other important suggestions you make are important to our articles. We followed your comments with the following modifications. Finally, thank you again for your comments and review, and have a pleasant life. The details of the modification are in the attachment

Author:Dr. Tingyu Wang、Mr. Lele Yan、Dr. Xinsheng Yang、Prof. Duyan Geng、Prof. Dr. Guizhi Xu *、Prof. Alan Wang *

  1. line 48: ‘make neurons depolarization’ should be ‘make neurons depolarize’

Thanks for your question about inappropriate use of parts of speech, we've made the following changes:

Original:

 which will make neurons depolarization and generate action potential, thus affecting the metabolism and electrical activity of nerves in the brain [4-7].

Revise:

which will make neurons depolarize and generate action potential, thus affecting the metabolism and electrical activity of nerves in the brain [4-7].

  1. line 102: the initial RTD should be defined at first use.

Thank you for your question about the fact that the RTD acronym is not explained in full when first proposed, and we have made the following additions in the text:

Original:

  1. Beatriz Goncalves et al present a single LED optrode neural tool, capable of assessing tissue temperature over time, and thermal imaging of brain tissue near the stimulation focus. RTD’s average accuracy of 0.2℃ at a normal body temperature of 37℃,and take into account the ability of electrical recording and light stimulation at the same time. The RTD thin film is integrated into the silicon carbon needle to adapt to light stimulation, electrophysiological recording point, temperature sensing, etc. This electrode has great potential in recording and stimulating neural activity[16].

Revise:

  1. Beatriz Goncalves et al. proposed a single LED phototube neural tool, which is internally integrated with RTD (resistance temperature detector) for sensing.It can evaluate tissue temperature over a period of time and perform thermal imaging of brain tissue near the stimulus focus. RTD’s average accuracy of 0.2℃ at a normal body temperature of 37℃, and take into account the ability of electrical recording and light stimulation at the same time. The RTD thin film is integrated into the silicon carbon needle to adapt to light stimulation, electrophysiological recording point, temperature sensing, etc. This electrode has great potential in recording and stimulating neural activity[16].

  1. line 362:  the grammar needs correction.

Thank you for your question about the syntax error of line 362. We have made the following modifications:

Original:

Figure 4. (A) Multi-target simulation results, all arrows indicate current direction (B) Schematic diagram of the arrangement of coils 1, 2, and 5 in the saline model (C) Simulation results of 1, 2, and 5 focus areas (D) ①, ② Time-domain distribution of induced electric field (E) Time-domain distribution of induced electric field at points ① and ③ (F) Time-domain distribution of induced electric field at point 4 (G) Schematic diagram of the arrangement of coils 1, 5 and 6 in the saline model (H) 1, 5, 6 focus area simulation results (I) Time-domain distribution of induced electric field at points ⑤, ⑥, ⑦.ouble-target simulation results (A), (B) Schematic diagram of the arrangement of coils 1, 2, and 5 in the saline model (C) Simulation results of 1, 2, and 5 focus areas (D) ①, ② Time-domain distribution of induced electric field (E) Time-domain distribution of induced electric field at points ① and ③ (F) Time-domain distribution of induced electric field at point 4 (G) Schematic diagram of the arrangement of coils 1, 5 and 6 in the saline model (H) 1 , 5, 6 focus area simulation results (I) Time-domain distribution of induced electric field at points ⑤, ⑥, ⑦.

Revise:

Figure 4. (A) Schematic diagram of the spatial positions of coils 1, 2, and 5 in a saline model, with arrows indicating the direction of coil current  (B)Schematic diagram of the arrangement side of coils 1, 2, and 5 in the saline water model (C)Simulation results of coils 1, 2, and 5 focus areas (D)Point ①, ② time-domain distribution of induced electric field  (E)Time-domain distribution of induced electric field at points ① and ③ (F)Time-domain distribution of induced electric field at point 4 (G) Schematic diagram of the spatial positions of coils 1, 5, and 6 in a saline model, with arrows indicating the direction of coil current (H) Simulation results of coils 1, 5, and 6 focus areas (I) Time-domain distribution of induced electric field at points ⑤, ⑥, ⑦.
